# Arthropod exosomal glycine-rich protein as a potential vaccine candidate effectively reduces tick blood-feeding and pathogen transmission

Waqas Ahmed[1,3], Wenshuo Zhou[2], Md Bayzid [iD][1], Denae Nadine LoBato[1], Kehinde D Fasae[1], Girish Neelakanta [iD][1] & Hameeda Sultana [iD][1✉]

## Abstract

During blood feeding, Ixodidae ticks secrete cement proteins, including glycine-rich proteins (GRPs), that facilitate attachment to the vertebrate host. However, the molecular mechanisms underlying exosomal GRP secretion at the feeding site and their roles in tick-pathogen interactions remain poorly understood. Here, we analyzed the *Ixodes scapularis* genome to identify salivary exosomal components involved in modulation of the tick-host skin interface. We identify an arthropod exosomal GRP (XM_002400035) that promotes transmission of Langat virus (LGTV), a tick-borne flavivirus, from ticks to vertebrate hosts. XM_002400035 was consistently upregulated in LGTV-infected *I. scapularis* ticks, tick-derived cells, and in tick exosomes. RNAi-mediated silencing of this exosomal GRP reduced viral loads, impaired tick blood-feeding efficiency, decreased tick body size and weights, and diminished LGTV acquisition and transmission. Similarly, active immunization of mice with recombinant GRP disrupted tick feeding, reduced tick fitness, and significantly impaired LGTV transmission from infected ticks to naive recipient hosts. Mechanistically, the exosomal GRP modulated host skin chemokine CXCL-12 levels at the feeding site. Together, these findings establish a dual role for a tick exosomal GRP in blood feeding and pathogen transmission and identify this tick exosomal GRP as a potential target for exosome-based transmission-blocking vaccines. More broadly, this work highlights arthropod exosomes as active mediators of flavivirus transmission and suggests new strategies for preventing and controlling tick-borne diseases.

**Keywords** Ticks & Flavivirus Transmission; Exosomal Glycine-rich Protein; Human Skin Chemokines; Wound-healing/repair; Immunization & Mice
**Subject Categories** Evolution & Ecology; Immunology

## Introduction

Ticks are blood-feeding arthropods that transmits microorganisms of major concern to human and animal health (Nuttall, 2023; Wikel, 2018). In recent years, arthropods have gained attention due to increase in cases with tick-borne diseases. *Ixodes scapularis* and *I. ricinus* are hard ticks and medically important vectors to study several tick-borne pathogens including neuroinvasive flaviviruses such as tick-borne encephalitis virus-TBEV, Powassan virus-POWV, Langat Virus-LGTV (a model pathogen similar to TBEV/POWV) and others (Brossard and Wikel, 2004; Hermance and Thangamani, 2018; Kazimírová et al, 2017; Neelakanta and Sultana, 2022; Nuttall and Labuda, 2003; Nuttall, 2023; Sultana and Neelakanta, 2020). Naive ticks acquire microorganisms during blood-feeding on infected vertebrate host and infected ticks feed on naive vertebrates to transmit pathogens. There are limited measures/ strategies available to prevent pathogen transmission and/or to treat vector-borne diseases. Therefore, detailed understanding of vector molecule(s) is essential to gain in-depth knowledge to understand the interactions with tick-borne pathogens.

Both acquisition/transmission of pathogens to or from *I. scapularis* ticks is mediated via saliva (secreted into feeding pit) (Kovar, 2004; Mulenga et al, 2022; Neelakanta and Sultana, 2022; Nuttall, 2023). Tick saliva is an intricate combination of non-peptidic and peptidic fluid secretion from salivary glands, that allows firm attachment on vertebrate host for a prolonged period of time (Grabowski et al, 2019; Mulenga et al, 2009; Neelakanta and Sultana, 2022; Nuttall, 2023). Most peptides and proteins secreted in Ixodid saliva are produced as tick feeding progresses (Grabowski et al, 2019; Neelakanta and Sultana, 2022). Tick saliva contains a plethora of components such as water, ions like Cl− and Na +, and secreted proteins like cement, serpins, anti-platelet aggregation factors, anti-coagulants, immunomodulators and anti-complement factors (Kotál et al, 2015; Nuttall, 2023). Recent study provided evidence that tick salivary glands and saliva from two different tick species (*I. scapularis* and *Amblyomma maculatum*) abundantly contained exosomes/extracellular vesicles (EVs) (Zhou et al, 2020). Tick feeding begins with secretion of saliva containing cement-like hardening substances that seal the gap between inserted mouthparts (hypostome and pair of chelicerae) and feeding lesions to be

[1]Department of Biomedical and Diagnostic Sciences, College of Veterinary Medicine, University of Tennessee, Knoxville, TN, USA. [2]China National Biotech Group (CNBG), CNBG-Virogin Biotech, Shanghai, China. [3]Present address: UNC HIV Cure Center, Institute of Global Health and Infectious Diseases, University of North Carolina at Chapel Hill, Chapel Hill, NC 27599, USA. ✉E-mail: hsultana@utk.edu

fixed into host skin (Bullard et al, 2016; Suppan et al, 2018; Vancová et al, 2020). Ticks can be easily pulled out of vertebrate skin before completion of cement secretion, but considerable force is needed after cement deposition. This indicates importance of cement in strengthening attachment of ticks' mouthparts onto host skin (Hollmann et al, 2018; Kim et al, 2014; Trimnell et al, 2005). Functions of cement cones have been well-studied (Lynn et al, 2022; Suppan et al, 2018; Villar et al, 2020). Tick feeding is divided into different stages of attachment, slow- and fast feeding processes (Karim et al, 2011; Lynn et al, 2022; Mulenga et al, 2009). Recent study identified and characterized tick cement in *Amblyomma americanum*, suggesting their tensile and adhesive characteristics and composition with pliable structures like silk (Hollmann et al, 2018). Tick cement protein (64TRP) of *Rhipicephalus appendiculatus* is an antigenic vaccine component that shows the ability to control tick-borne diseases (Havlíková et al, 2009; Labuda et al, 2006). Observations of tick cementome in *R. microplus* suggested that ticks merge host and tick-derived proteins to synergize in cement formation, facilitate attachment, solidification, modulation of host immune responses, feeding and detachment (Villar et al, 2020). In addition, role of tick cement/cone is well-studied in perspective of feeding in *A. americanum* (Bullard et al, 2016; Hollmann et al, 2018), *R. appendiculatus* (Bishop et al, 2002), and *Haemaphysalis spinigera* ticks (Chinery, 1973).

Arthropods have evolved strategies to evade and counterattack host defenses (Karim et al, 2011; Mulenga et al, 2009; Neelakanta and Sultana, 2022; Nuttall, 2023; Ribeiro and Francischetti, 2003). Potential role of tick saliva is highlighted for pathogen transmission via exosomes (Hackenberg and Kotsyfakis, 2018; Sultana and Neelakanta, 2020; Zhou et al, 2020). Our previous studies showed that infectious exosomes enable transmission of flaviviruses from arthropod to mammalian cells (Hackenberg and Kotsyfakis, 2018; Regmi et al, 2020; Sultana and Neelakanta, 2020; Vora et al, 2018; Zhou et al, 2020; Zhou et al, 2018; Zhou et al, 2019). We are the first to report the presence of flaviviral full-length RNA genomes, viral transcripts (both negative and positive strands) and Non-structural (NS1)/envelope-(E) proteins/polyproteins inside exosomes-derived from mosquito, tick, mouse and human cells (Vora et al, 2018; Zhou et al, 2018). We have reported the detection of full-length dengue (DENV)-RNA genome in infected mosquito cell-derived exosomes that facilitates its transmission from arthropod to mammalian cells (Vora et al, 2018). In addition, extensive evidence supports that sphingomyelinase-like enzyme (*Is*SMase) from *I. scapularis* plays a key role in membrane-associated viral replication and exosome biogenesis (Regmi et al, 2020). This study revealed insights in contribution of vector defense mechanism(s) against tick-borne viral infections and antiviral pathways (Regmi et al, 2020). Exosomal marker HSP70 (Heat-Shock Protein-70)-like-protein is reduced upon feeding on immunodeficient animals suggesting its important role in fibrinogen lysis (Vora et al, 2017). Also, HSP70 is detected in tick exosomes indicating salivary factors secretion via exosomes (Zhou et al, 2018). Exosome are reported as carriers of bioactive salivary factors that enable pathogen transmission and successful blood-feeding through regulating the immune response at tick-human skin interface (Zhou et al, 2020). Several studies defined contributions of arthropod exosomes in viral entry, cargo delivery, regulation of immune response against viral pathogens, and exosome-mediated pathogen transmission from ticks and mosquitoes (Ahmed et al, 2021; Butler et al, 2023; Chávez et al, 2021; Hackenberg and Kotsyfakis, 2018; Neelakanta and Sultana, 2022; Sultana and

Neelakanta, 2020; Urbanelli et al, 2019; Van Dongen et al, 2016; Zhou et al, 2020; Zhou et al, 2018). However, in-depth understanding of molecular repertoires and their functions in vector-pathogen interactions via exosomes is yet unknown. The central objective of this study is to identify the tick salivary exosomal cargo that may facilitate the delay in wound closure, and repair process noted in human skin keratinocytes (HaCaT cells) treated with tick exosomes derived from saliva, salivary glands, or tick ISE6 cells (Zhou et al, 2020). We believe that identification and targeting of tick salivary exosomal molecules in pathogen transmission is highly important. We also hypothesize that tick salivary exosomes may facilitate cement-like glycine-rich proteins (GRPs) secretion during blood-feeding. Our current study reveals an essential role of exosomal GRP in mediating-tick-borne virus transmission and in facilitating blood-feeding. We show that exosomal GRP affects pathogen acquisition and transmission, exosome biogenesis, and modulates host immune evasion via regulating human chemokine CXCL-12. This study not only investigates contribution of GRP molecules in tick-virus-host interactions but also highlights the important role of arthropod exosomes and exosomal content in facilitating tick feeding and pathogen transmission.

## Results

### Bioinformatic and comparative analysis of *Ixodes scapularis* glycine-rich protein sequences

We performed bioinformatics, prediction, and comparative analyses with primary amino acid sequences of *I. scapularis* cement-like glycine-rich proteins (GRPs) downloaded from National Center for Biotechnology Information (NCBI). Detailed information of GRPs with their respective GenBank, nucleotide/vector base accessions and signal peptides is shown (Appendix Table S1). The amino acid sequence from GenBank accession number EEC20123.1 (referred with nucleotide accession number XM_002400035 or as exosomal glycine-rich protein in this study) is aligned with amino acid sequences of other GRPs by using ClustalW alignment program (DNASTAR Lasergene). Alignment analysis revealed some degree of conservation in amino acid sequences of GRP (Appendix Fig. S1). We also noted that GRP family of proteins are glycine-rich in residues from 185 to 272 (Appendix Fig. S1). Sequence identity revealed that molecule-XM_002400035 ranged from low (11.11%) to high (46.3%) with other *I. scapularis* GRPs (Appendix Fig. S2A). Phylogenetic analysis showed that XM_002400035 falls within same clade with EEC18488.1. Other GRPs formed different clades (Appendix Fig. S2B). Furthermore, comparative bioinformatic analysis of *I. scapularis* GRPs was performed with *Rhipicephalus microplus* (GenBank acc. No. JAC59038.1, JAC59109.1), *R. appendiculatus* (GenBank acc. No. AAK98794.1, BAG56007.1), *Amblyomma americanum* (GenBank acc. No. ACG76243.1, JAG92708.1), *A. variegatum* (GenBank acc. No. DAA34670.1) and *Hyalomma excavatum* (GenBank acc. No. JAP66598.1) ticks (Appendix Fig. S3). Number of GRPs amino acids (Appendix Fig. S3A), transmembrane domains (Appendix Fig. S3B), Protein Kinase C (PKC) phospho-sites (Appendix Fig. S3C), N-glycosylation-sites (Appendix Fig. S3D), N-myristoylation-sites (Appendix Fig. S3E), and casein kinase II phospho-sites (Appendix Fig. S3F) in *I. scapularis* and other ticks is analyzed at PROSITE (Appendix Fig. S3). Protein feature

Expression of glycine-rich family genes during *Ixodes scapularis* tick blood feeding

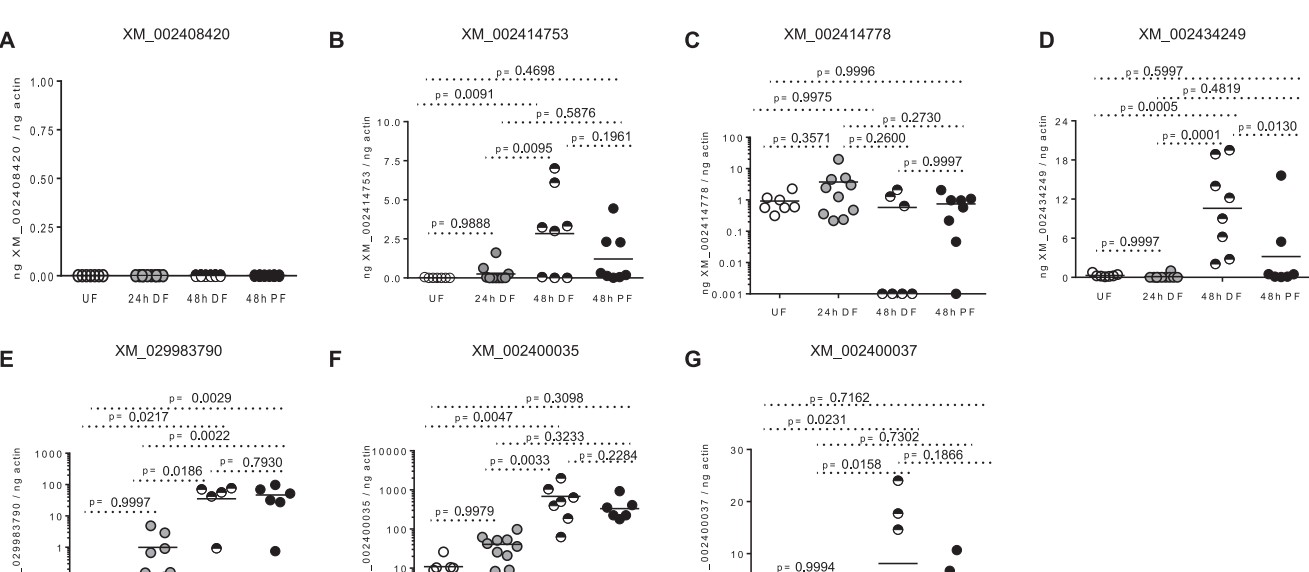

**Figure 1. Expression of GRP molecules in unfed/fed *Ixodes scapularis* uninfected nymphal ticks.**

(A–G) Seven genes- XM_002408420 (A), XM_002414753 (B), XM_002414778 (C), XM_002434249 (D), XM_029983790 (E), XM_002400035 (F), and XM_002400037 (G) expression in uninfected unfed (UF), and 24 h during-feeding (DF), 48 h DF/48 h post-fed (PF) ticks is shown by QRT-PCR analysis. Each circle denotes one tick. Exact sample numbers for each panel representing multiple experiments are UF (7 ticks in (A–G)), 24 h DF (7 ticks in (A), and 10 ticks in (B–G)), 48 h- DF/PF (8 ticks in (A–D) and (G), 7 ticks in (E) and (F)). The mRNA expression levels of GRP molecules are normalized to tick beta-actin mRNA levels. Statistical differences were calculated using one-way ANOVA with Tukey's multiple comparisons test and p values are shown. p < 0.05 is considered as statistically significant. Source data are available online for this figure.

prediction analysis revealed presence of 2 transmembrane domains and 10 N-myristoylation-sites, however, no PKC phospho-sites, N-glycosylation and casein kinase II phospho-sites were observed in EEC20123 GRP (Appendix Fig. S3).

## Expression analysis of *Ixodes scapularis* GRPs genes during tick blood-feeding

We first amplified ten GRP genes (1–10, Appendix Fig. S4 and Appendix Table S2) from both ticks and tick cells by using following oligonucleotides (Appendix Table S2). We noted that out of 10 selected genes, only 7 are highly expressed in ISE6 tick cells, therefore we proceeded only with these molecules. We analyzed glycine-rich gene expression in uninfected *I. scapularis* ticks (unfed/fed for 24 h and 48 h during-feeding (DF)/48 h post-feeding PF). QRT-PCR analysis revealed no differences for XM_002408420 (Fig. 1A, with barely detectable transcript levels) and XM_002414778 (Fig. 1C). However, significant differences in expression levels of XM_002414753 (Fig. 1B), XM_002434249 (Fig. 1D), XM_029983790 (Fig. 1E), XM_002400035 (Fig. 1F) and XM_002400037 (Fig. 1G) were noted between unfed/ 24 h/48 h-DF or 48 h-PF ticks. We observed highest expression levels of XM_029983790, XM_002400035 and XM_002400037 at 48 h-DF/ PF, whereas XM_002414753 and XM_002434249 had higher expression at 48 h-DF but lower at 48 h-PF that are like expression levels observed at 24 h-DF. Expression of XM_002400035 was noted to be significantly higher in all feeding timepoints (of 24 h/48 h-DF)/post-

fed (48 h-PF) ticks when compared to levels in unfed ticks (Fig. 1F). These results show that XM_002400035 is differentially expressed in *I. scapularis* ticks.

## Tick-borne LGTV modulates *GRPs* genes expression in *Ixodes scapularis* ticks

To determine whether LGTV has any impact on glycine-rich gene expressions, infected ticks were generated by in vitro synchronous method (Mitzel et al, 2007; Taank et al, 2018). QRT-PCR analysis revealed significantly (P < 0.05) higher viral burden in LGTV-infected unfed/partially-fed (24 h-DF) *I. scapularis* nymphal ticks in comparison to their respective uninfected controls (Fig. 2A). Next, we determined expression of glycine-rich genes in unfed/24 h-partially-fed ticks infected with LGTV (Fig. 2B–H). QRT-PCR analysis showed barely detectable/very-low transcripts of XM_002408420 (Fig. 2B). No significant differences in expression of XM_002414753 were noted (Fig. 2C). We found significant (P < 0.05) downregulation in expression levels of XM_002434249 (Fig. 2E) between LGTV-infected unfed and 24 h-partially-fed ticks but no differences were found in other groups. Also, no significant differences were noted in expression of XM_002414778 (Fig. 2D), XM_029983790 (Fig. 2F), and XM_002400037 (Fig. 2H). It is interesting to note that XM_002400035 was significantly (P < 0.05) upregulated during partial-feeding in uninfected and LGTV-infected ticks when compared to uninfected and LGTV-infected

Tick-borne Langat virus regulates *I. scapularis* glycine-rich family genes during blood feeding

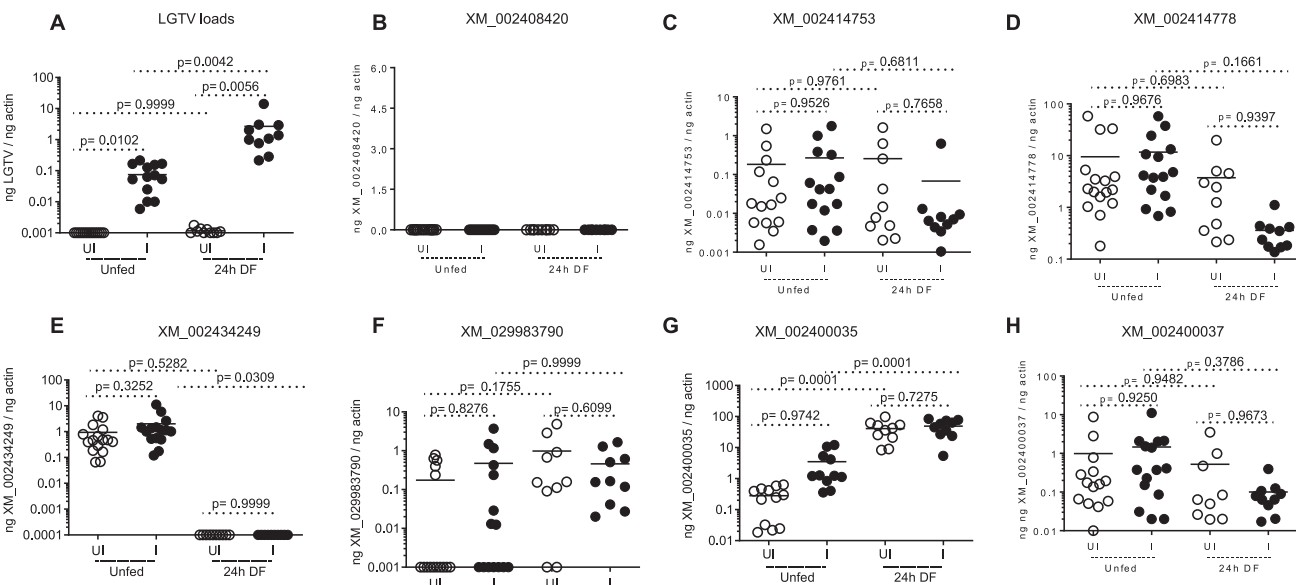

**Figure 2. LGTV-infection significantly modulates GRP expression in unfed/fed ticks.**

QRT-PCR analysis showing LGTV load (**A**) and expression of seven transcripts (**B–H**): XM_002408420 (**B**), XM_002414753 (**C**), XM_002414778 (**D**), XM_002434249 (**E**), XM_029983790 (**F**), XM_002400035 (**G**) and XM_002400037 (**H**) in unfed/24 h-fed uninfected (UI) or synchronously-generated LGTV-infected (I) ticks. Each circle denotes one tick. UI 24 h DF values were considered from Fig. 1 for comparison to I-24 h DF. Exact sample numbers for each panel representing multiple experiments are UI-UF (15 ticks in (**A**), (**B**) and (**F**); 14 in (**C**) and (**H**); 16 in (**D**, **E**); and 12 ticks in (**G**)), I-UF (15 ticks in (**A**), (**B**), (**D–F**) and (**H**); 14 in (**C**) and 11 in (**G**)), UI-24 h DF and I-24 h PF (10 ticks in (**A–H**)). LGTV-loads and transcript levels of GRPs are normalized to tick beta-actin levels. Statistical differences were calculated using one-way ANOVA with Tukey's multiple comparisons test and $p$ values are shown. $p < 0.05$ is considered as statistically significant. Source data are available online for this figure.

unfed ticks suggesting partial-feeding specifically induces expression of XM_002400035 in both uninfected and LGTV-infected ticks (Fig. 2G).

## LGTV-infection selectively upregulates expression of XM_002400035 in tick cells/tick cell-derived exosomes and enhances exosome secretion

LGTV readily infects tick cells with higher viral loads detected at 72 h post-infection (p.i.) (Zhou et al, 2018). This timepoint is efficient for collection of exosomes from tick cells (Zhou et al, 2018). QRT-PCR analysis revealed increased LGTV-loads at 72 h p.i when compared to early timepoint of 24 h p.i. (Fig. 3A; Appendix Fig. S5A). We detected LGTV-loads in tick cell-derived exosomes at 72 h p.i. (Fig. 3B). Next, we investigated all GRPs transcripts upon LGTV-infection in tick cells. QRT-PCR analysis showed no significant differences ($P > 0.05$) in expression levels of XM_002408420 (Appendix Fig. S5B), XM_002414753 (Appendix Fig. S5C), XM_002414778 (Appendix Fig. S5D), XM_002434249 (Appendix Fig. S5E), XM_029983790 (Appendix Fig. S5F) and XM_002400037 (Appendix Fig. S5H) between uninfected/LGTV-infected (24 h/72 h p.i.) tick cells. However, at 72 h p.i., XM_002400035 was significantly ($P < 0.05$) upregulated in LGTV-infected tick cells when compared to levels in uninfected controls (Fig. 3C; Appendix Fig. S5G). No differences were found in transcript levels of XM_002400035 at 24 h p.i., suggesting that XM_

002400035 is perhaps required at later timepoints (Appendix Fig. S5G). We could not detect other transcripts (with several attempts) in tick cell-derived exosomes, except XM_002400035. We found that XM_002400035 transcripts were significantly ($P < 0.05$) upregulated in LGTV-infected (72 h p.i.) exosomes when compared to uninfected controls (Fig. 3D).

As XM_002400035 is highly detected in LGTV-infectious exosomes, we determined the exosome numbers by Spectradyne's nCS1™ particle analyzer and by diluting exosomal samples with PBS/Tween 20 (PBST) in 1:2000 ratio. Particle analysis revealed that tick cell-derived exosomes (measured as particle number against concentration of particles), were more in number at size ranging between 50 and 110 nm (Fig. 3E,F; Appendix Fig. S6A,B). Particle concentration and number count measurements revealed that exosomes derived from uninfected tick cells were less in concentration ($1.23 \times 10^9 \pm (4.38 \times 10^7, 5.56 \times 10^8)$/ml, $2.69 \times 10^9 \pm (2.04 \times 10^8, 1.52 \times 10^8)$/ml) (Fig. 3G; Appendix Fig. S6C) and numbers ($N = 596 \pm 186$, and $N = 345 \pm 28$) when compared to exosomes from LGTV-infected group (Fig. 3H; Appendix Fig. S6D). Concentration of exosomes-derived from LGTV-infected cells were noted to be $3.51 \times 10^9 \pm (4.41 \times 10^7, 5.49 \times 10^7)$/ml, and $5.63 \times 10^9 \pm (3.13 \times 10^8, 2.41 \times 10^8)$/ml, and particle numbers were ($N = 1504 \pm 212$, and $N = 583 \pm 98$) (Fig. 3G,H; Appendix Fig. S6C,D). These findings highlight that LGTV not only induces expression of XM_002400035 but also increases secretion of exosomes from tick cells.

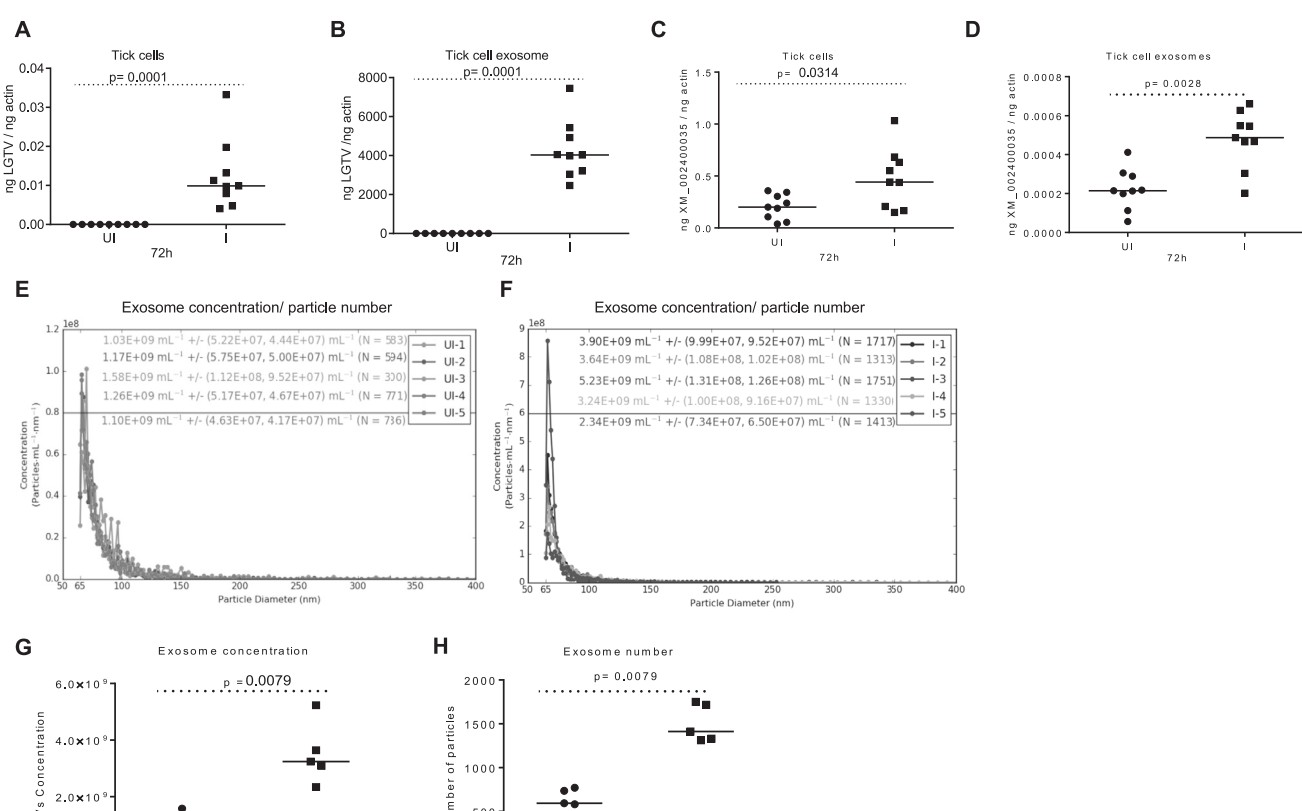

Figure 3.   **LGTV-infection upregulates expression of XM_002400035 in *Ixodes scapularis* tick cells, and tick cell-derived exosomes.**

QRT-PCR analysis showing LGTV-loads (**A**, **B**, MOI 1) or XM_002400035 mRNA expression levels (**C**, **D**) in tick cells/exosomes, respectively. LGTV-loads and mRNA expression levels of XM_002400035 were normalized to tick beta-actin mRNA levels. Circles denote uninfected (UI) group, whereas squares represent LGTV-infected group. Each circle/square denotes data value from independent culture well. Spectradyne nCS1 nanoparticle analyzer counted particle sizes of exosomes collected from uninfected (**E**) or LGTV-infected (**F**) tick cells using TS-400 filter cartridge. (**G**) shows EVs concentration and particles numbers (**H**) measured from uninfected (UI)/ LGTV-infected (I) groups, respectively. Exact sample numbers for each panel representing multiple experiments are UI/I group (9 replicates in (**A–D**); and 5 replicates in (**E–H**)). Statistical differences were calculated using Mann–Whitney U test and p values are shown. p < 0.05 is considered as statistically significant. Source data are available online for this figure.

## Expression of XM_002400035 is developmentally regulated and induced upon LGTV-infection of larval and nymphal ticks

We examined whether exosomal XM_002400035 expression is modulated during *I. scapularis* life cycle and developmental stages. QRT-PCR analysis showed that XM_002400035 is developmentally regulated in *I. scapularis* unfed uninfected ticks. Larvae and nymphs showed significantly higher expression (*P* < 0.05) of XM_002400035, when compared to levels noted in adult ticks (male/female) (Fig. 4A). However, comparison of XM_002400035 transcripts between larvae and nymphs or between male/female ticks revealed no significant differences (*P* > 0.05) (Fig. 4A). Furthermore, we generated fed-larval ticks by feeding larvae on uninfected/LGTV-infected mice. QRT-PCR analysis revealed viral load detection in all individually processed larval ticks (Fig. 4B). We noted that transcripts of XM_002400035 were significantly

(*P* < 0.05) increased in LGTV-infected larval ticks in comparison to levels in uninfected ticks (Fig. 4C). A group of fully-fed larvae (120 h-PF) repleted from LGTV-infected/uninfected mice were allowed to molt into nymphs. QRT-PCR analysis of these molted nymphs revealed detection of viral loads in all individually molted unfed ticks (Fig. 4D). Furthermore, we noted that XM_002400035 transcripts were significantly higher (*P* < 0.05) in LGTV-infected-unfed nymphs in comparison to uninfected ticks (Fig. 4E). These studies suggest that XM_002400035 is modulated in different *I. scapularis* life cycle/developmental stages, in fed larvae and during transstadial transmission of LGTV from larval to nymphal ticks.

## Expression of XM_002400035 is consistently induced upon LGTV-acquisition and transmission

In addition to molted nymphs, we generated 48 h-DF/120 h-PF nymphal ticks through acquisition experiment by allowing naive

*I. scapularis* novel exosomal glycine-rich gene XM_002400035 is upregulated upon LGTV acquisition and transmission

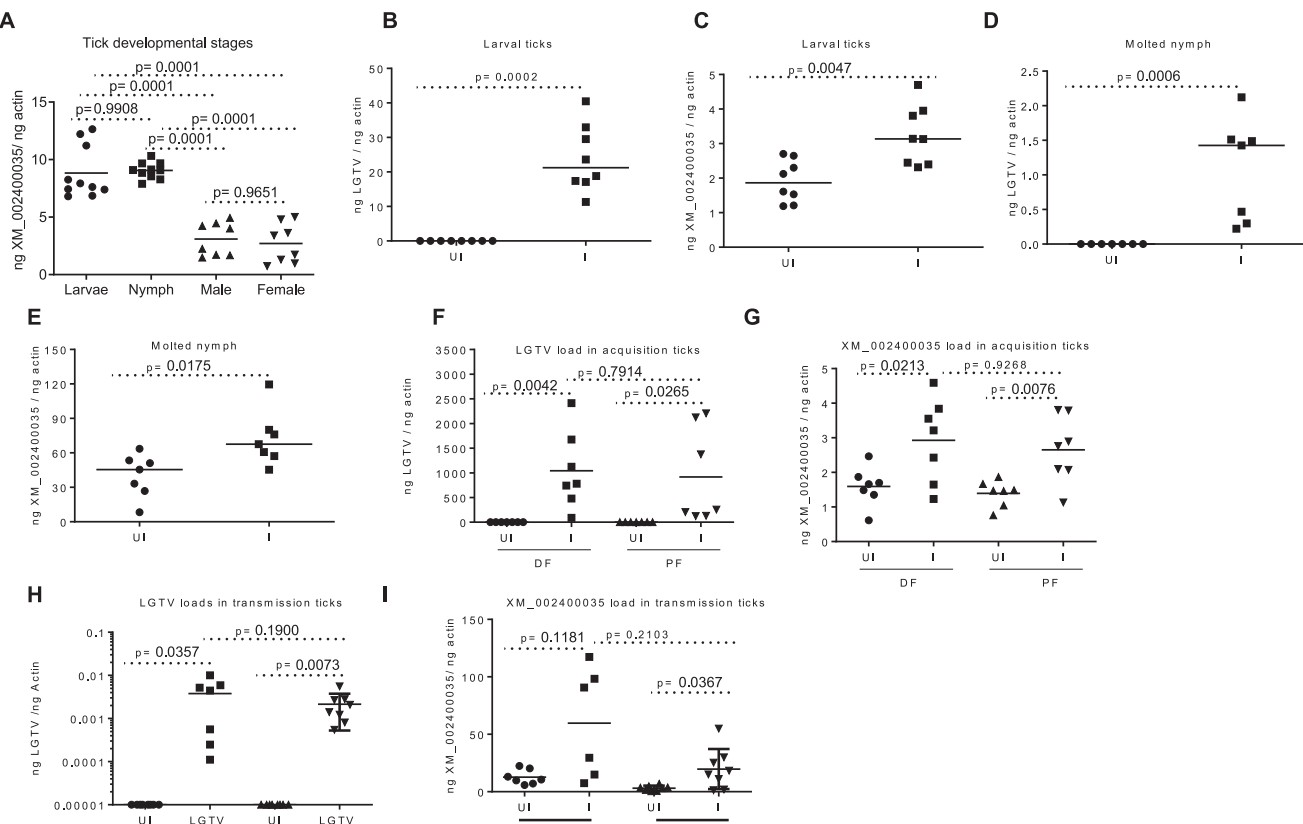

**Figure 4. Expression of XM_002400035 in developmental stages of *Ixodes scapularis* ticks and during-feeding/post-fed ticks.**

QRT-PCR analysis showing XM_002400035 expression in developmental stages of unfed-uninfected larval (circles), nymphs (squares) or adult (male and female; triangles/inverted triangles, respectively) ticks (**A**). LGTV-loads (**B**), and XM_002400035 expression in fed-uninfected (circles) or LGTV-infected (squares) larval ticks (**C**), LGTV-loads (**D**) and XM_002400035 expression in unfed uninfected (circles)/LGTV-infected (squares) nymphs (**E**), LGTV-loads (**F**) and XM_002400035 expression in 48 h during-feeding (DF) (uninfected-circles, and infected-squares) and 48 h post-fed (PF) (uninfected-triangles and infected-inverted triangles) (**G**) ticks generated from acquisition of LGTV from mice to ticks is shown. LGTV-loads (**H**) and XM_002400035 expression in 48 h DF and 48 h PF (**I**) ticks generated from transmission of LGTV from infected ticks to mice is shown. Each circle/square/triangle/inverted-triangle denotes one tick. Exact sample numbers for each panel representing multiple experiments is: (**A**) Larvae (10 replicates, with pool of 5 larvae as each replicate), Nymphs (10), adult male and female (8 ticks in each group), (**B, C**) (8 ticks in UI/I groups), (**D, E**) (7 ticks in each group), (**F, G**) (7 ticks in each groups UI-DF, I-DF, UI-PF, I-PF), (**H**) (8 in UI-DF, 7 in I-DF, 8 in each UI-PF and I-PF) and (**I**) (7 ticks in UI-DF, 6 in I-DF, 8 in each UI-PF and I-PF). XM_002400035 mRNA levels or LGTV-loads were normalized to tick beta-actin levels. Statistical differences in (**A**) and (**F–I**) were calculated using one-way ANOVA with Tukey's multiple comparisons test, and in (**B–E**), we used Mann–Whitney U test and p values are shown. $p < 0.05$ is considered as statistically significant. Source data are available online for this figure.

ticks to feed completely on LGTV-infected/uninfected mice. Higher viral loads were detected in ticks collected during 48 h-DF/120 h-PF timepoints that were fed on LGTV-infected mice (Fig. 4F). QRT-PCR analysis showed that XM_002400035 transcripts were significantly increased in LGTV-infected nymphs collected from either during-feeding (48 h-DF) or post-feeding (120 h-PF) timepoints when compared to levels noted in respective uninfected controls (Fig. 4G). For transmission studies, a batch of fully-fed larvae (120 h-PF) repleted from independent batch of LGTV-infected/uninfected mice were allowed to molt into nymphs. These unfed-uninfected/LGTV-infected nymphs were fed on uninfected mice. QRT-PCR analysis showed higher viral loads in LGTV-infected nymphs (Fig. 4H). Also, significantly increased XM_002400035 transcripts were evident in LGTV-infected nymphs collected from post-feeding

(120 h-PF) when compared to levels noted in respective uninfected controls (Fig. 4I). These data suggest that exosomal GRP expression is upregulated upon LGTV-acquisition (from infected mice to naive ticks) or LGTV-transmission (from infected ticks to naive mice) to facilitate pathogen transport.

## RNA interference of XM_002400035 impacts its expression and LGTV-loads in tick cells/exosomes and in nymphal ticks during pathogen acquisition

Details of dsRNA generation and transfections followed by LGTV-infection are described in Methods (Appendix Fig. S7A,B). No morphological changes were noted in tick cells transfected with XM_002400035-dsRNA in comparison to mock-dsRNA-treated control (empty L4440 vector) (Appendix Fig. S7C). The empty

Silencing of novel exosomal glycine-rich gene affects LGTV loads in tick cells and during acquisition from mice into ticks

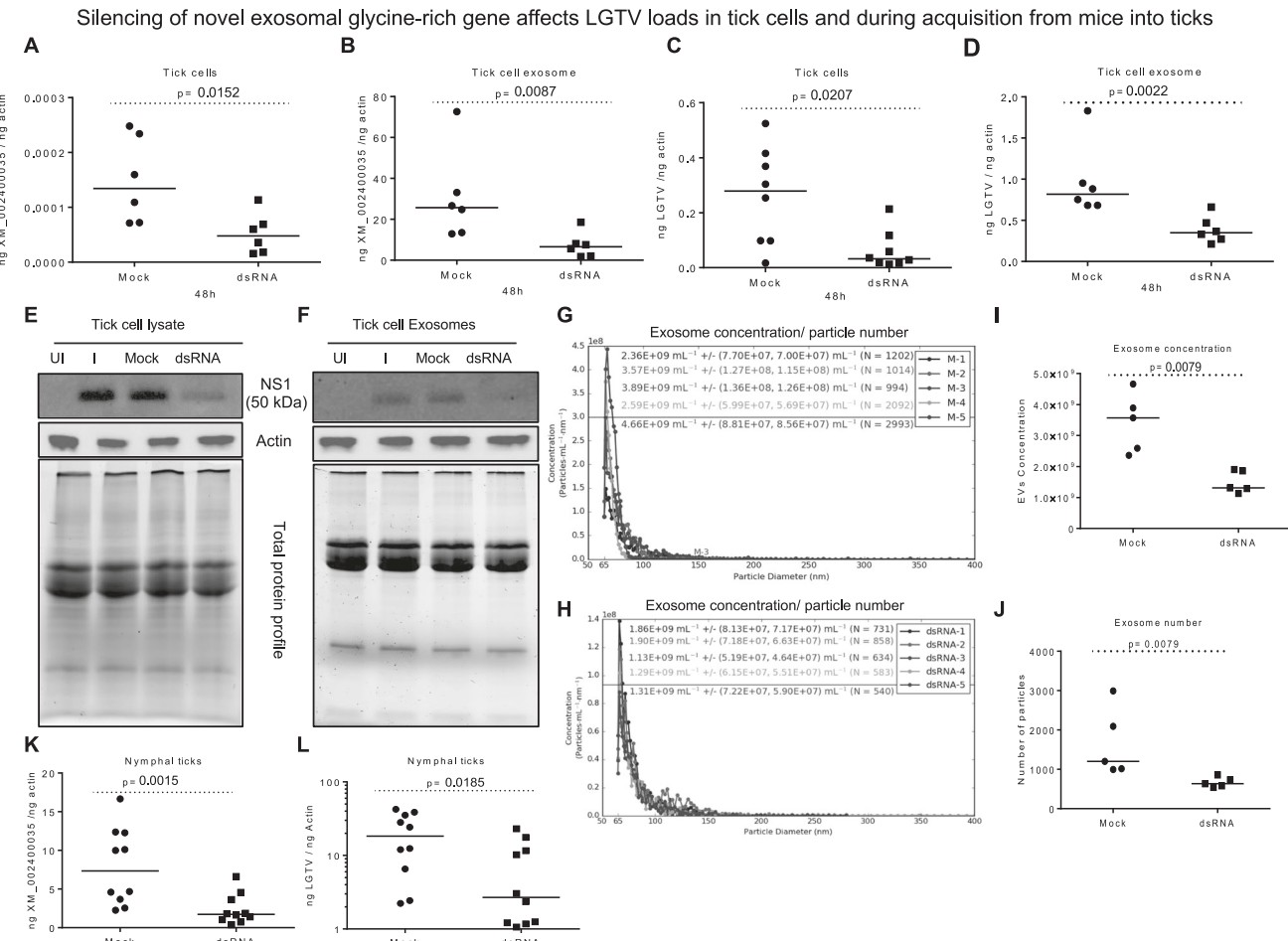

**Figure 5. RNAi-mediated silencing of XM_002400035 affects LGTV burden in tick cells, tick cell-derived exosomes and in nymphs during pathogen acquisition.**

QRT-PCR analysis showing exosomal XM_002400035 expression (A, B) or LGTV-loads (1 MOI, 48 h p.i.) (C, D) to reveal silencing efficiency (72 h post-transfection) in tick cells (A)/exosomes (B), respectively. Immunoblotting analysis showing detection of LGTV Non-structural protein 1 (NS1-protein; 50 kilodalton/kDa) in total tick cell lysates (E)/exosomes (F) prepared from uninfected (UI)/LGTV-infected (I), mock-dsRNA/XM_002400035-dsRNA-treated cells at 72 h p.i. Total protein profiles are stain-free gel images (loading controls). (G) and (H) showing particle sizes and counts using TS-400 filter cartridge (65–400 nm). (I) shows extracellular vesicle (EV) concentration and number of exosome particles (J) in tick cell-derived exosomes collected from mock-dsRNA/XM_002400035-dsRNA-treated groups. QRT-PCR analysis showing XM_002400035 mRNA expression levels (K)/LGTV burden (L) to reveal silencing efficiency and viral loads in repleted ticks microinjected with mock-dsRNA/XM_002400035-dsRNA-treated groups. Circles denote mock-dsRNA-treated groups, and squares represent XM_002400035-dsRNA-treated groups (A–D and K, L). Each circle/square indicates one tick. Exact sample numbers for each panel representing multiple experiments are (A), (B), (D) (6 replicates), and (C) (8 replicates) in both mock/dsRNA groups, (G–J) (5 replicates in mock/XM_002400035-dsRNA-treated groups), and in (K, L) (10 ticks in each group). XM_002400035 mRNA levels/LGTV-loads were normalized to tick beta-actin levels. Statistical differences were calculated using Mann–Whitney U test and p values are shown. p < 0.05 is considered as statistically significant. Source data are available online for this figure.

plasmid pL4440 is a non-tick-related dsRNA that serves as an appropriate control for XM_002400035-dsRNA group. Silencing of exosomal GRP revealed significant ($P < 0.05$) decrease in XM_002400035 transcripts in tick cells and in exosomes when compared to respective mock-dsRNA-treated controls (Fig. 5A,B). These data suggest successful silencing efficiency of XM_002400035 in tick cells that perhaps eventually affect its packaging in exosomes (Fig. 5A,B). We noted that upon XM_002400035-dsRNA treatment, LGTV-loads were significantly ($P < 0.05$) reduced in both tick cells (Fig. 5C) and in exosomes (Fig. 5D) when compared to LGTV-loads in respective mock-dsRNA-treated controls. Immunoblotting analysis showed higher non-structural protein 1 (NS1-protein, ~50 kDa) loads in LGTV-infected and mock-dsRNA-treated groups

in comparison to LGTV-infected and XM_002400035-dsRNA-treated tick cells/exosomal total lysates (Fig. 5E,F). Presence of NS1 further confirmed that exosomes contained LGTV proteins that are reduced upon silencing of exosomal GRP (Fig. 5E,F). As expected, we detected similar LGTV-loads in infected tick cells and mock-dsRNA-treated controls. Actin loads and total profile gel images served as loading controls (Fig. 5E,F). The quantification of these immunoblots (Fig. 5E,F) is shown in Appendix Table S3. These data suggest that silencing of exosomal GRP affects LGTV-loads in tick cells and viral transmission via reduced exosomes secretion from these cells. Particle measurements confirmed increased number of exosomes in size ranges of 50–110 nm (Fig. 5G,H; Appendix Fig. S8A,B). Increased exosome concentration (3.41 ×

$10^9 \pm (5.03 \times 10^7, 4.73 \times 10^7)$/ml, $4.91 \times 10^9 \pm (2.48 \times 10^8, 1.97 \times 10^8)$/ml) (Fig. 5G,I; Appendix Fig. S8C,D) and particle numbers ($N = 1659 \pm 870$, and $678 \pm 103$) (Fig. 5G,I,J; Appendix Fig. S8D) were noted in mock-treated controls when compared to concentrations ($1.50 \times 10^9 \pm (6.77 \times 10^7, 5.97 \times 10^7)$/ml, $2.65 \times 10^9 \pm (2.52 \times 10^8, 1.70 \times 10^8)$/ml) and particle numbers ($N = 669 \pm 127$ and $268 \pm 168$) of exosomes in XM_002400035-dsRNA-treated tick cells (Fig. 5H–J; Appendix Fig. S8C,D). Reduced LGTV-loads directly corelated with lower exosomes concentrations and numbers noted upon silencing of exosomal GRP. These data suggested that XM_002400035 is directly involved in tick exosomal biogenesis pathway(s) and in LGTV-transmission from cells to exosomes.

To generate XM_002400035 knockdown in nymphs, we directly injected dsRNA into body of naive/uninfected ticks. Microinjection methods are previously described (Taank et al, 2018). After 24 h recovery from microinjection, XM_002400035 knockdown nymphs were fed for 120 h-PF on LGTV-infected mice. QRT-PCR analysis showed a significant ($P < 0.05$) reduction in exosomal GRP transcripts in XM_002400035-dsRNA-treated ticks in comparison to mock-dsRNA-treated ticks (Fig. 5K). In addition, we found significantly ($P < 0.05$) reduced LGTV-loads in XM_002400035-dsRNA-treated ticks in comparison to mock-dsRNA-treated controls (Fig. 5L). These findings further highlight the significance of exosomal GRP in acquisition of tick-borne LGTV from infected vertebrate host to naive ticks.

## Silencing of XM_002400035 affects LGTV-transmission from infected ticks to naive vertebrate host

We next examined whether lower numbers and concentrations of tick exosomes from LGTV-infected-XM_002400035-deficient ticks could influence viral transmission from infected ticks to mice. *I. scapularis* ticks need only one blood meal per stage (as larva, nymph/adult stage), indicating need for transstadial transmission of pathogen upon the following blood meal. LGTV-infected unfed nymphs were generated by molting of larvae. These LGTV-infected nymphs were microinjected with mock-dsRNA/XM_002400035-dsRNA to determine the effect of pathogen transmission from vector to vertebrate host. After 24 h recovery from microinjections, LGTV-infected and mock-dsRNA/XM_002400035-dsRNA-treated ticks were fed on uninfected mice. Repleted ticks were collected and analyzed for engorgement/body weights. Micrographs showed that LGTV-infected and XM_002400035-dsRNA-treated ticks were dramatically reduced in body sizes when compared to sizes noted with mock-dsRNA-treated controls (Fig. 6A). In addition, significant lower body weights were recorded in XM_002400035-dsRNA-treated ticks when compared to mock-dsRNA-treated ticks (Fig. 6B). We noted ~2-fold reduction in tick body weights of XM_002400035-dsRNA-treated ticks when compared to the mock-dsRNA-treated ticks (Fig. 6B). These data implicate that reduced body weights is likely a result of decrease blood-feeding by these XM_002400035-knockdown ticks. QRT-PCR analysis confirmed silencing efficiency with significantly ($P < 0.05$) decreased expression in XM_002400035-dsRNA-treated ticks when compared to mock-dsRNA-treated ticks (Fig. 6C). We also noted significantly ($P < 0.05$) reduced LGTV-loads in XM_002400035-dsRNA-treated ticks in comparison to mock-dsRNA-treated ticks (Fig. 6D). In addition, lower NS1 loads were noted in XM_002400035-dsRNA-treated ticks when compared to levels in mock-dsRNA-treated

controls (Fig. 6E). Total protein profile gel image serves as loading control (Fig. 6E). The quantification of this immunoblot (Fig. 6E) is shown in Appendix Table S3.

Next, we examined LGTV-transmission from infected ticks to naive mice. Analysis of mouse blood showed that all naive mice were infected on day 7 post tick placement/attachment (Fig. 6F). However, reduced viral loads were detected in blood of infected mice that allowed feeding of LGTV-infected-XM_002400035-dsRNA-treated ticks when compared to control group of mice that allowed feeding of LGTV-infected-mock-dsRNA-treated ticks (Fig. 6F). We also observed reduced viral loads from mice tissues (brains and spleens) that allowed feeding of LGTV-infected XM_002400035-dsRNA-treated ticks when compared to levels noted in control mice (Fig. 6G). We also noted that CXCL-12 transcripts were significantly ($P < 0.05$) increased in skin tissue of these mice that allowed feeding of LGTV-infected XM_002400035-dsRNA-treatd ticks when compared to levels observed in skins of control mice (Fig. 6H). Collectively, our results show that exosomal GRP is critical not only for tick blood-feeding but also for LGTV-transmission from ticks to vertebrate host.

## Immunization with exosomal glycine-rich protein impairs LGTV-transmission from infected ticks to naive vertebrate host

We next examined whether active immunization of mice with purified recombinant GST-glycine-rich protein (GRP)/EEC20123.1 exosomal protein could influence LGTV-transmission from infected ticks to naive mice. Detailed cloning, expression and purification of exosomal GRP protein is shown (Appendix Fig. S9A–C). The fragment of EEC20123.1 exosomal protein (35 amino acids polypeptide or 105 bp PCR fragment, denoted as highlighted region) cloned into pGEX-6P-2 expression vector is shown (Appendix Fig. S9A). Restriction enzyme digested product and *Escherichia coli* colonies screening for confirmation are shown (Appendix Fig. S9B,C). The GST-GRP induction and expression are shown (Appendix Fig. S10A). Coomassie-blue-stained gel image shows purified GST/GST-GRP (Fig. 7A). Experimental details of mice immunization are detailed in Methods (Appendix Fig. S10B). Enzyme-linked immunosorbent assay (ELISA) and immunoblotting analysis showed that GST-GRP purified protein and total lysates from adult tick salivary glands/SG-derived-EVs significantly ($P < 0.05$) bind to serum collected from GST-GRP immunized mice in comparison to the serum from GST-protein immunized mice (in 1:200/1:100 sera dilutions, respectively) (Fig. 7B,C; Appendix Fig. S11A). In addition, immunoblots and ELISA assay further confirmed the binding of GST-GRP-immunized mice serum (antibodies) and showed reduced levels of endogenous GRP in total lysates prepared from LGTV-infected-mock-dsRNA/XM_002400035-dsRNA-treated tick cells/fed nymphs (Fig. 7C; Appendix Fig. S11B). Total protein profile gel images serve as loading control (Fig. 7C). The quantification of these immunoblots (Fig. 7C) is shown in Appendix Table S3. These data suggest that serum raised against GST-GRP had higher binding affinity and recognizes its endogenous protein.

To determine the effect of pathogen transmission from vector to vertebrate host, we fully fed LGTV-infected nymphs on mice immunized with GST/GST-GRP. Repleted ticks (48 h PF) were collected, and body sizes/weights were analyzed. Micrographs

Silencing of novel exosomal glycine-rich gene affects LGTV loads during transmission from ticks into mice

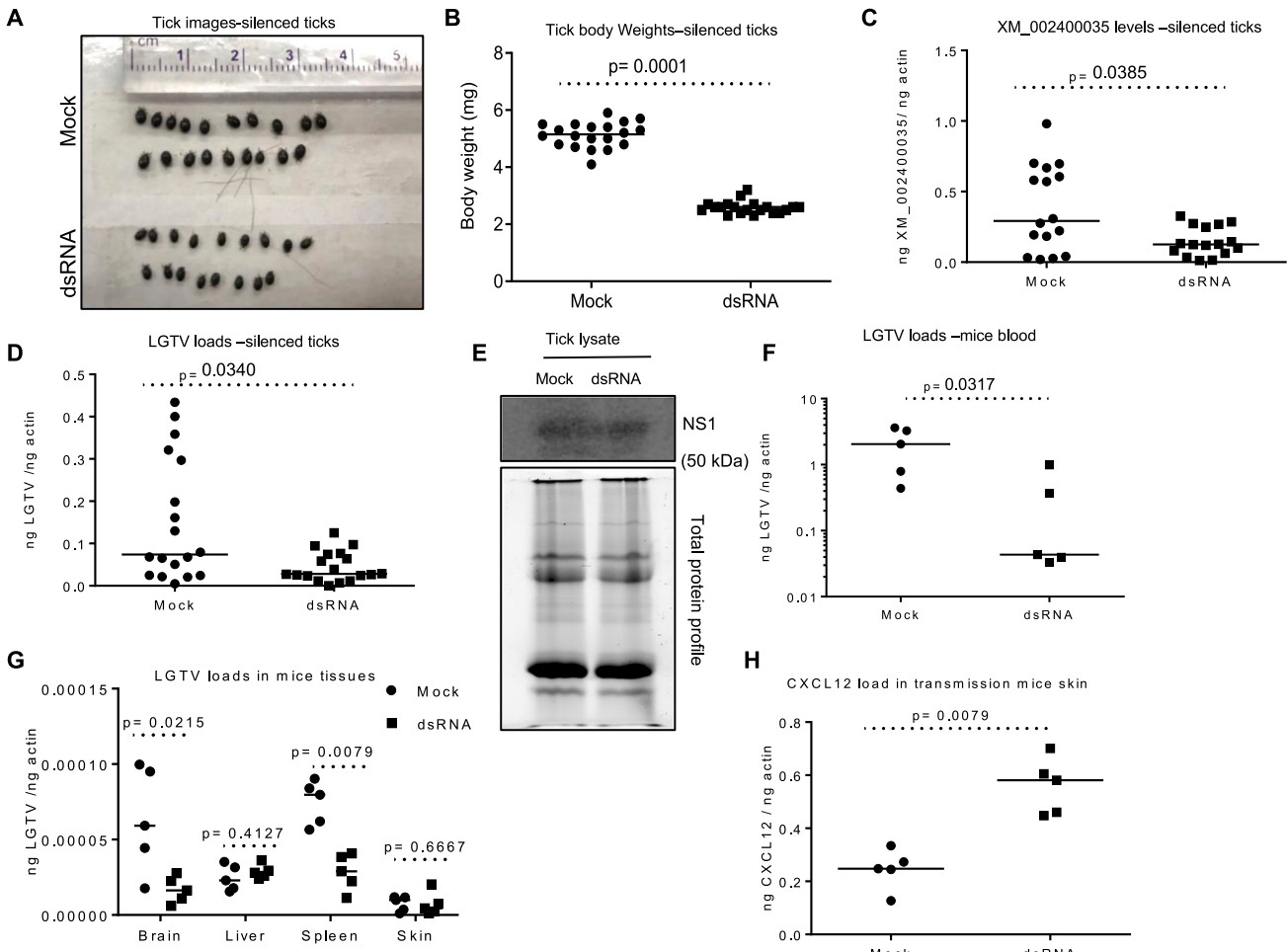

**Figure 6. Silencing of exosomal XM_002400035 affects LGTV-transmission from infected-ticks to naive mice.**

LGTV-infected ticks were microinjected with mock-dsRNA/ XM_002400035-dsRNA to determine silencing efficiency and LGTV-loads. Repleted ticks were immediately aligned on double-sided sticky tapes. Images were taken from mock-dsRNA/ XM_002400035-dsRNA-microinjected ticks (**A**). A group of repleted ticks was also immediately weighed using microbalance and tick body weights are shown (**B**). QRT-PCR analysis showing XM_002400035 expression (**C**) and LGTV-loads (**D**), in mock-dsRNA/XM_002400035-dsRNA-microinjected nymphal ticks generated from transmission of LGTV from infected ticks to mice. Circles denote mock-dsRNA-treated groups, and squares represent XM_002400035-dsRNA-treated groups (**B–D**). Each circle/square indicates one tick. Exact sample numbers for each panel representing multiple experiments is (**B**) (20 ticks in both mock/dsRNA-treated groups), (**C**) (16 ticks) and 6D (18 ticks) in mock/XM_002400035-dsRNA-treated groups. (**E**) Immunoblotting analysis showing levels of NS1 protein (50 kilodalton/kDa) in mock-dsRNA/XM_002400035-dsRNA-microinjected nymphal ticks generated from transmission of LGTV from infected ticks to mice. Total protein profile is a stain-free gel image (loading control). LGTV-loads were detected in mouse blood (**F**), and tissues (**G**) of all infected mice that allowed feeding of mock-dsRNA/XM_002400035-dsRNA-treated LGTV-infected ticks. Circles denote samples generated from mice allowing feeding of mock-dsRNA-treated groups, and squares represent samples generated from mice allowing feeding of XM_002400035-dsRNA-treated groups (**F, G**). Each circle/square indicates one mouse (**F, G**). XM_002400035 mRNA levels/LGTV-loads were normalized to tick (**C, D**) or mouse (**F, G**) beta-actin levels. (**H**) CXCL-12 expression analyzed in skin of mice that allowed feeding of mock-dsRNA/XM_002400035-dsRNA-treated nymphal ticks is shown. Circles denote mock-dsRNA-treated group, and squares represent XM_002400035-dsRNA-treated group. In (**F–H**), 5 independent mice in each group is shown. Statistical differences in all panels were calculated using Mann–Whitney U test and *p* values are shown. In (**G**), statistical comparisons for viral loads in different tissues were made only between mice allowing feeding of either XM_002400035-dsRNA-treated ticks/mouse allowing feeding of mock-dsRNA-treated ticks. No multiple comparison between tissues is performed. The *p* < 0.05 is considered as statistically significant. Source data are available online for this figure.

showed that LGTV-infected ticks fed on GST-GRP immunized mice were dramatically reduced in their body sizes with significantly (*P* < 0.05) lower body weights when compared to sizes/weights noted for LGTV-infected ticks fed on GST-protein immunized mice (Fig. 7D,E). We noted ~1.6-fold reduction in body weights of ticks fed on GST-GRP immunized mice when compared to the GST-protein immunized mice (Fig. 7E). Reduced body

weights directly correlated with decreased blood meal intake by these ticks (Fig. 7E). QRT-PCR analysis confirmed significant (*P* < 0.05) decreased in XM_002400035 transcripts in ticks fed on GST-GRP immunized mice when compared to levels noted in ticks fed on GST-protein immunized mice (Fig. 7F). We noted significantly (*P* < 0.05) reduced LGTV-loads in ticks fed on GST-GRP immunized mice when compared to levels observed in control

Immunization with novel exosomal glycine-rich protein affects LGTV loads during transmission from ticks into mice

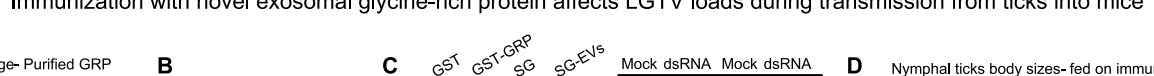

### Figure 7. Immunization with an exosomal GRP affects LGTV-transmission from infected ticks to naive mice.

(A) Coomassie stained gel image (full) showing purified GST-GRP/GST-protein from *E. coli* BL-21 cell lysates. ELISA (B) and immunoblot (C) performed with purified proteins (GST/GST-GRP)/total lysates from adult tick-SGs/SGs-derived-EVs and probed with serum collected from GST/GST-GRP immunized mice (1:200 or 1:100 for panels (B)/(C), respectively) are shown. Immunoblots performed with GRP immunized mice serum are shown for detection of GRP levels in adult tick salivary glands or EVs-derived from adult tick salivary glands or total tick cell lysates or total nymphal tick lysates prepared from LGTV-infected mock-dsRNA/XM_002400035-dsRNA-treated groups (C). After repletion, fully engorged ticks (fed on GST/GST-GRP proteins immunized mice) were immediately aligned on double-sided tapes. Images were taken from ticks that fed on GST/GST-GRP immunized mice (D). A group of repleted ticks was immediately weighed using microbalance and tick body weights are shown (E). QRT-PCR analysis showing XM_002400035 expression (F) and LGTV-loads (G) in ticks that fed on GST/GST-GRP immunized mice. Micrograph showing tick body sizes (H) and reduced molting rates (I) of adult ticks that were collected as repleted nymphs fed on GST/GST-GRP immunized mice. QRT-PCR-loads detected in molted adult ticks that had fed as nymphs on GST/GST-GRP immunized mice before molting (J). ELISA assay showing presence of GRP antibodies in tick lysates obtained from repleted nymphs/adult ticks generated by molting (K). LGTV-loads were also detected in blood (L), and tissues (M) of GST/GST-GRP protein immunized mice that allowed feeding of LGTV-infected ticks. Circles denote GST group, and squares represent GST-GRP group. Each circle/square indicates data generated from 5 to 6 replicates. Exact sample numbers for each panel representing multiple experiments in (B, L, M) is 5 mice in GST/6 mice in GST-GRP groups), (D, E) (22 ticks in GST and 21 in GST-GRP group), (F, G) (12 ticks in each group) and (H–J) (17 ticks in GST and 10 in GST-GRP group), and (K) (8 ticks in each group). In (K), showing ELISA, we had 6 replicates in duplicate for each group. In (M), statistical comparisons in different tissues were made only between mice that allowed feeding of XM_002400035-dsRNA-treated ticks or mock-dsRNA-treated ticks. No multiple comparison between tissues was performed. Statistical differences in all panels were calculated using Mann–Whitney U test and p values are shown. p < 0.05 is considered as statistically significant. Source data are available online for this figure.

ticks fed on GST-protein immunized mice (Fig. 7G). Furthermore, we allowed molting of these repleted nymphs into adult ticks. Micrographs showed that adult ticks (nymphs that fed on GST-GRP immunized mice) were reduced in their body sizes when compared to sizes noted for adult ticks (nymphs that were fed on GST-protein immunized mice) (Fig. 7H). Also, we noted that rate of molting efficiency of these nymphs into adult ticks was reduced (Fig. 7I). It is interesting to note that LGTV-loads in adult ticks

were also significantly (*P* < 0.05) reduced in comparison to control ticks (Fig. 7J). We did not find any significant differences in XM_002400035 transcript levels in these adult ticks (Appendix Fig. S11C). Significantly reduced LGTV-loads in both fed nymphs and molted adult ticks revealed continuous blocking of exosomal GRP by the antibodies taken up by the ticks during blood feeding on GST-GRP-immunized mice in comparison to the control group of ticks fed on GST-immunized mice (Fig. 7K). We performed

ELISA assay and found higher binding of GST-GRP purified protein to tick protein lysates collected from repleted nymphs and adult ticks that were fed on GST-GRP immunized mice when compared to lysates of nymphs and adult ticks that fed on GST-protein immunized mice (Fig. 7K). Next, we examined LGTV-transmission from infected-ticks to naive mice. QRT-PCR analysis of LGTV-loads in blood showed that all naive mice were infected on day 7 post tick placement/attachment (Fig. 7L). Significantly ($P < 0.05$) reduced viral loads were detected in blood and tissues such as brains, spleens and skins collected from GST-GRP immunized mice when compared to GST-protein immunized mice (Fig. 7L,M), suggesting exosomal GRP as anti-tick vaccine candidate.

Since, exogenous treatment of GST-GRP showed delay in wound healing and repair process (Fig. 8A), we addressed if anti-GRP sera obtained from immunized mice could affect the GRP induced wound closure delay. We performed a wound healing assay with HaCaT cells and considered untreated (UT) group of cells as internal control (Fig. 8A). We noted that GRP treatment and similarly GRP and GST-protein antisera (obtained from GST-protein immunized mice) treated HaCaT cells showed delay in wound closure and repair process (Fig. 8A). It was noteworthy that treatment with GRP antisera (obtained from GST-GRP immunized mice) closed wounds and repaired the monolayer of HaCaT cells (Fig. 8A). We noted, a rapid closure in wound with repair process in this group of HaCaT cells treated with GRP antisera (Fig. 8A). Measurement of wound diameter showed similar results (Fig. 8B). HaCaT cells with tick GRP, and wound scratch (for 72 h), showed remaining wound size percentages in following order: 5.6% (for GRP alone), 9.3% (for GRP and GST-protein immunized sera), 0% (for GRP and GST-GRP immunized serum, respectively). The untreated (UT) group showed 0% in wound size percentage and served as control (Fig. 8B). Like HaCaT in vitro cell line, we found that primary cultures of keratinocytes from normal human adult skin cells showed very similar pattern in wound healing assay performed with untreated (UT) group of cells as internal control (Appendix Fig. S12A). We found that GRP alone treatment showed a delay in wound closure and repair process (Appendix Fig. S12A). It was interesting to note that treatment with GRP along with GST-GRP antisera (obtained from GST-GRP immunized mice) closed wounds and repaired the scratch in human keratinocytes (Appendix Fig. S12A). This wound closure was comparable to the untreated group (Appendix Fig. S12A). Measurement of wound diameter showed similar results (Appendix Fig. S12B). Keratinocytes with tick GRP, and wound scratch (for 72 h), showed remaining wound size percentages in the following order: 53.1% (for GRP alone), 25.3% (for GRP and GST-GRP immunized serum, respectively). The untreated (UT) group showed 16.5% in wound size percentage and served as control (Appendix Fig. S12B). This data suggest that antisera against tick GRP exosomal protein is a potential candidate in controlling the delay in wound healing and repair process during tick bites.

### Silencing of exosomal GRP reduced inflammation at tick bite site in mice and exogenous GST-CXCL-12 treatment closed wounds in human keratinocytes

To evaluate inflammation at tick bite site, LGTV-infected-mock-dsRNA-treated/LGTV-infected-XM_002400035-dsRNA-treated

ticks were allowed to feed on uninfected mice. We observed increased inflammation in mice skin that allowed feeding of LGTV-infected-mock-dsRNA-treated ticks. Presence of panniculus with moderate to large number of neutrophils, lymphocytes, plasma cells, and few macrophages were observed in these mice (Fig. 9A; Appendix Fig. S13A). Moderate number of inflammatory cells with lymphocytes mixed with few neutrophils, macrophages and plasma cells were observed in mice skins that allowed feeding of LGTV-infected-XM_002400035-dsRNA-treated ticks (Fig. 9B; Appendix Fig. S13B). Higher magnification images are shown for greater details (Appendix Fig. S13A,B). ELISA assays with GST-GRP immunized mice serum determined significantly ($P < 0.05$) decreased binding (1:1000 dilution) to skin lysates collected from mice that allowed feeding of LGTV-infected-XM_002400035-dsRNA-treated ticks when compared to binding noted with control group (Fig. 9C). Next, we determined, if HaCaT cells treatment with human CXCL-12 (2 μg protein fused with GST)/GST-GRP/GST-alone-protein in combination with tick exosomes (collected from uninfected/LGTV-infected tick cells) has any effect(s) on tick exosome-mediated delay in wound-healing and repair. It was noteworthy that treatment (for 24 h) with CXCL-12-protein enhanced wound closures and repair when compared to GST-GRP (Fig. 9D). Untreated group served as control and showed wound closure and repair process as expected (Fig. 9D). HaCaT-cells treated with GST-protein and exosomes from uninfected tick cells showed closure of wound gap, however, we noted a delay in wound closure and repair process with GST-protein group and exosomes from LGTV-infected group (Fig. 9D). GST-GRP group with exosomes from both uninfected and LGTV-infected tick cells failed to close wounds and showed gaps (Fig. 9D). Measurement of wound diameter showed similar results (Fig. 9E). HaCaT cells with CXCL-12-protein, LGTV-infectious exosomes and wound scratch (for 24 h), showed remaining wound size percentages in following order: 7.2%/15.5% (uninfected/LGTV-infected-exosomes-treated with GST-protein, respectively), 14.3%/18.4% (uninfected/LGTV-infected-exosomes-treated with GST-GRP, respectively), 0% (LGTV-infected and GST-CXCL-12-protein-treated), or 0% (untreated control) (Fig. 9E). GST-CXCL-12-protein and exosomes from uninfected ticks cells are shown in previous study (Zhou et al, 2020). No significant differences were noted with GST-protein group (Fig. 9D,E). Also, we noted that CXCL-12 transcripts were significantly ($P < 0.05$) increased in skins of GST-GRP immunized mice when compared to levels observed in skins from GST-protein immunized mice (Fig. 9F). These data suggest that LGTV-transmission from infected ticks to GST-GRP immunized mice skin is affected due to increased CXCL-12-protein production.

## Discussion

During blood-feeding, transmission of tick-borne pathogens occurs from infected arthropod to the vertebrate host (Kazimírová et al, 2017; Nuttall, 2023). Cement is a substance secreted in Ixodidae saliva that aids in anchorage of tick mouthparts to host skin in order to facilitate feeding for longer times and to defend from being groomed off by the host (Suppan et al, 2018; Villar et al, 2020). Ticks produce feeding lesion and inhibits vertebrate host immune responses to take complete blood meal, while pathogens manipulate host and tick biological processes to facilitate their transmission

Antisera against tick exosomal glycine-rich protein corrects delays in cell migration, repair and wound closures

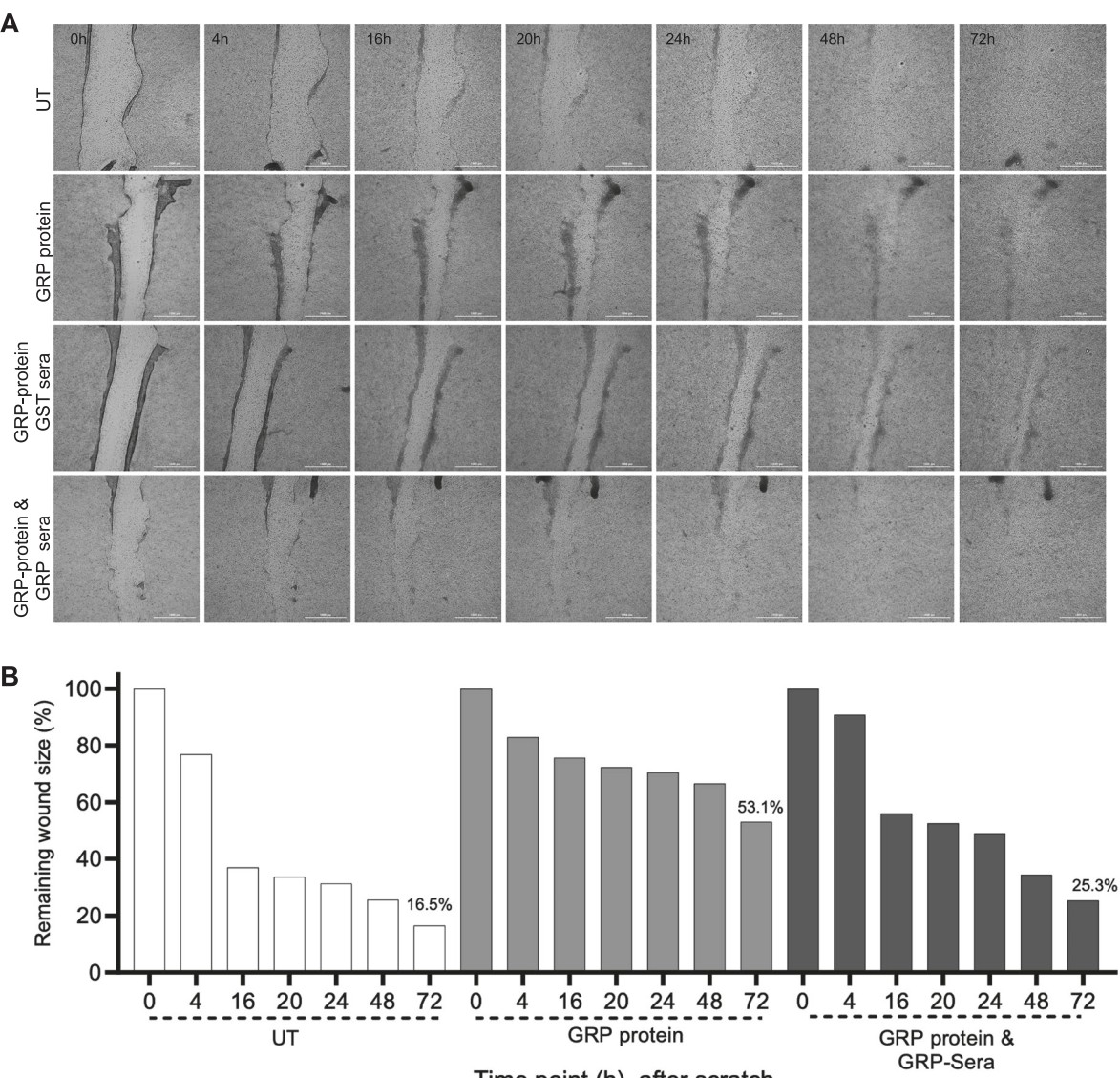

**Figure 8. Treatment with GST-GRP antisera restored wound-healing and repair process in skin keratinocytes.**

(A) Scratch-based assays performed on HaCaT cell monolayers incubated with 2 µg/ml of GST-GRP protein alone (for 72 h), or GST-GRP with GST-protein immunized mice antisera or GST-GRP protein with GST-GRP immunized mice sera in 1:100 dilution are shown. Phase contrast images (obtained using Cytation 7 imaging system) of HaCaT cell monolayers were taken for selected time-points (of 0, 4, 16, 20, 24, 48, and 72 h) and using 10× magnification. Untreated (UT) monolayers of HaCaT cells served as internal control. Scale bar indicates 1000 µm for each image per group/timepoint. (B) Measurement of remaining wound size diameters (analyzed by ImageJ software) at different time-points (of 0, 4, 16, 20, 24, 48, and 72 h) post-treatment are shown. Wounds at 0 h is considered as 100% for all groups, including untreated (UT) control. Experiment was replicated for multiple times in the laboratory. In (B), statistical differences in all panels were calculated using Mann–Whitney U test and p values are shown. $p < 0.05$ is considered as statistically significant. Source data are available online for this figure.

(Lynn et al, 2022; Nuttall, 2023). Blocking tick cement function during blood-feeding could prevent pathogen transmission (Bowman and Nuttall, 2008; Kazimírová et al, 2017; Lynn et al, 2022; Nuttall, 2023). There are several evidences that highlight tick cement cones in counteracting against host immune response(s) (Kazimírová and Stibraniova, 2013; Nuttall, 2023). Recent advancement has emphasized functional involvement of tick exosomes in pathogen transmission (Hackenberg and Kotsyfakis, 2018; Sultana and Neelakanta, 2020). Our studies showed that tick cell-derived

exosomes play key roles in tick-borne flaviviruses transmission from arthropod to vertebrates and evades host skin immune response (Chávez et al, 2021; Vora et al, 2018; Zhou et al, 2020; Zhou et al, 2018). To our knowledge, this is the first report highlighting role of an important exosomal cement-like glycine-rich protein (GRP) in transmission of a tick-borne flavivirus via arthropod exosomes. We now showed role of exosomal GRP in modulating host immune responses at tick bite site to allow/ease and mediate blood-feeding process. Combing the *I. scapularis*

Silencing of novel exosomal glycine-rich protein reduced inflammation at tick bite site and treatment with human GST-CXCL12 closed wounds

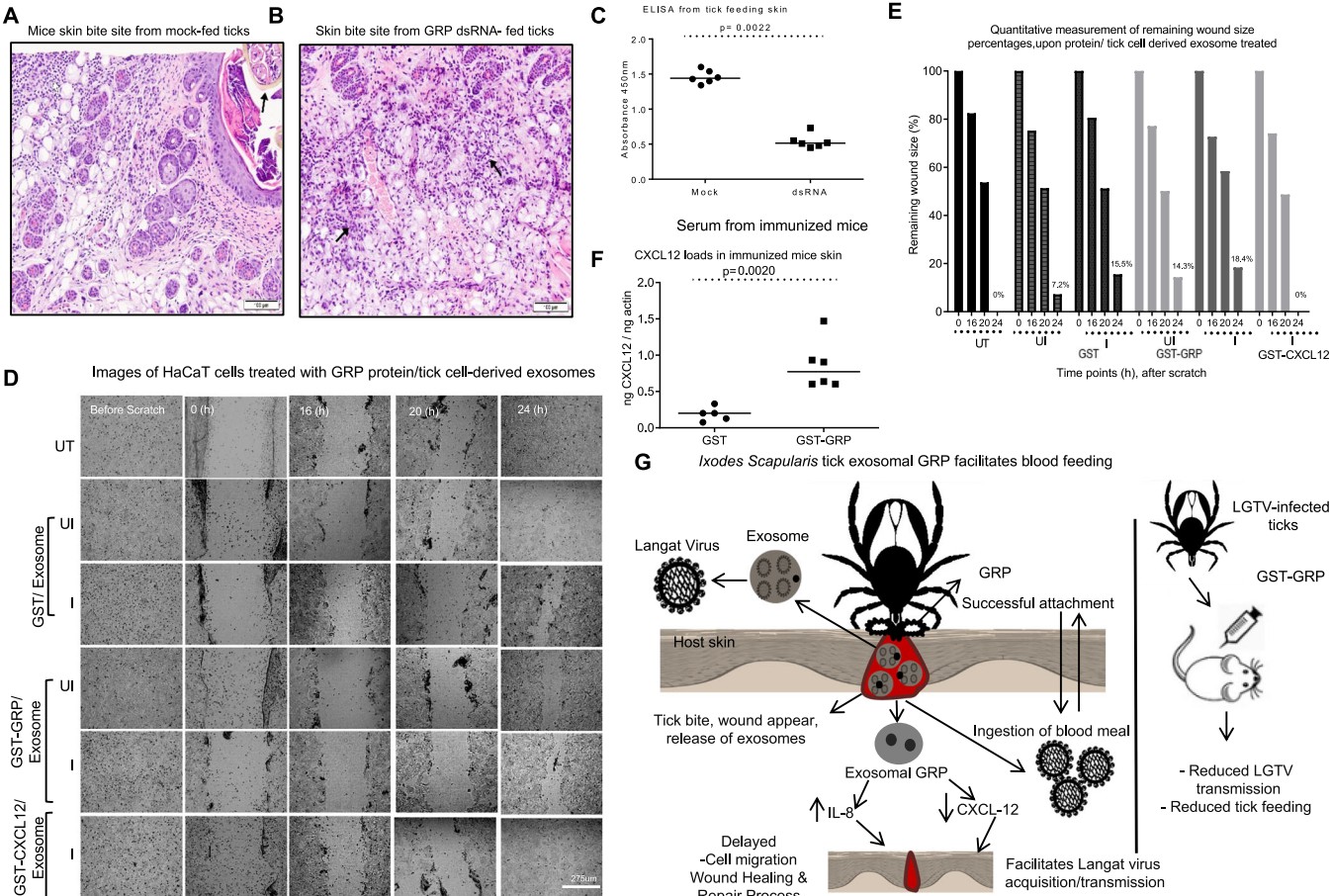

**Figure 9.  Silencing of tick exosomal GRP reduces inflammation at bite site in mice and treatment with GST-CXCL-12 protein restored wound-healing and repair process in skin keratinocytes.**

Hematoxylin and eosin (H&E) staining of skin biopsy samples from mice (that allowed feeding of mock/XM_002400035-dsRNA-treated ticks) display that tick feeding causes inflammation at bite site in mock-dsRNA-treated group (**A**), but inflammation is reduced in XM_002400035-dsRNA-treated group (**B**). Within the panniculus, there is downward projection of epidermis containing chitinous tick mouthparts (shown by black arrow). The panniculus contained moderate to large number of neutrophils, lymphocytes, plasma cells, and lower number of macrophages in mice that allowed feeding of mock-dsRNA-treated ticks (**A**). However, panniculus contained moderate number of inflammatory cells (shown by black arrow) with lymphocytes mixed with few macrophages and plasma cells in mice that allowed feeding of XM_002400035-dsRNA-treated ticks. There is a mild crush artifact in this image. Magnification of both these images is 200×. Scale bar indicates 100 μm for each image. Enlarged images shown in Fig. 9A,B are repeated in Appendix Fig. S13A,B for better visualization. (**C**) ELISA assay performed with skin lysates from mice that allowed feeding of ticks silenced for exosomal GRP or mock control ticks. Samples were probed with serum from immunized mice (1:1000 dilution). (**D**) Scratch assays performed on HaCaT cell monolayers incubated with 2 μg of GST/GST- GRP/GST-CXCL-12 protein (for 12 h), with/without 20 μl of tick exosomes from uninfected (UI), LGTV-infected (I), LGTV-infected and mock-dsRNA-treated or LGTV-infected and XM_002400035-dsRNA-treated groups are shown. Phase contrast images (obtained using EVOS auto-fluorescence system, M7000) of HaCaT cell monolayers were taken for selected time-points (as before scratch, 0, 16, 20, and 24 h) and using 10× magnification. Untreated (UT) monolayers served as internal control. Scale bar indicates 275 μm for each image per group or timepoint. (**E**) Measurement of remaining wound size diameters (analyzed by ImageJ software) at different time-points (of 0, 16, 20, and 24 h) post-treatment of tick exosomes-derived from UI, I, mock/XM_002400035-dsRNA is shown. Wounds at 0 h were considered as 100% for all groups, including untreated (UT) control. Mouse CXCL-12 expression was analyzed in skin samples from mice immunized with GST/GST-GRP protein is shown (**F**). Exact number of sample numbers for each group representing multiple experiments is 5 mice for GST/6 mice for GST-GRP groups (in **C**, **F**). Statistical differences were calculated using Mann–Whitney U test and p value is shown. p < 0.05 is considered as statistically significant. (**G**) Schematic model showing tick-borne flavivirus transmission to vertebrate host via tick saliva-derived exosomes. *Ixodes scapularis* tick attaches firmly and bites on host skin for longer feeding. Secreted saliva contains a plethora of substances including cement and perhaps cement-like GRPs to seal the feeding cone/cavity for directional blood flow and to defend from being groomed off by the vertebrate host. During blood meal ingestion, infected-ticks may continuously spit saliva containing infectious exosomes with viral full-length RNA genomes or polyproteins at host skin interface. We propose that incubation of tick exosomes containing exosomal GRP modulates the battle ground at skin interface by delaying cell migration/recruitment of immune cells (like neutrophils and dendritic cells from circulation) at the wound/bite site. Tick exosomes containing GRP inhibits residential keratinocytes and IL-8/CXCL-12 to delay injury, wound-healing, tissue damage, and repair process that will eventually enable ticks to acquire a successful blood meal at the host skin interface. Source data are available online for this figure.

genome revealed several cement-like glycine-rich proteins. We considered only 10 GRPs that showed complete CDS sequences. This information is the only rationale for these gene selection and for further analysis. Partial sequences were not considered due to risk of redundancy in sequence information. In addition, the central objective of this study aimed at identification and characterization of tick salivary exosomal molecule(s) in vector-pathogen interactions. The data available from NCBI and VectorBase resources clearly suggested these genes/proteins accession numbers as a group of cement molecules. However, we used a neutral or descriptive nomenclature based on their primary sequence feature of glycine-rich and refer them in this study as glycine-rich proteins (GRPs). The cement family of proteins are known for the formation of cement cones during tick feeding (de Souza et al, 2024; Hollmann et al, 2018; Mulenga et al, 2022; Simo et al, 2017). The role of these GRP molecules and in particular the exosomal GRP in cement cone formation is our future perspective and is not in the scope of this current study.

Detailed comparative bioinformatic analysis showed a high degree of amino acid sequence identity between *I. scapularis* GRPs. Furthermore, phylogenetic evaluation disclosed that *I. scapularis* GRPs align in different clades. Prediction as GRP from *R. appendiculatus*, *I. scapularis* and *H. longicornis* showed close relatedness with one another. Presence/absence of functional motifs like transmembrane domain numbers, PKC/CK2 phosphorylation/N-Glycosylation/N-Myristoylation-sites revealed differences between GRPs from *I. scapularis*. These analyses provide important information on domain organization and supports future studies on these family of molecules. Out of ten selected GRPs, three of the transcripts (XM_002400840, XM_002404575 and XM_002414656) had lower amplification in tick cells and hence were not further analyzed. Expression analysis of 7 transcripts during tick blood-feeding showed very low/basal levels in unfed ticks. Out of seven, only five transcripts showed induced expression during 48 h-DF/48 h-PF. Among the five induced transcripts during blood-feeding, only XM_002400035 showed highest expression in 24 h/48 h-DF and 48 h-PF ticks. It is interesting to note that XM_002400035 is mostly regulated during blood-feeding (at 24 h/48 h-DF), otherwise they are kept at low levels in unfed ticks. Our expression analysis (24 h-DF/48 h-DF/48 h-PF) further strengthens that XM_002400035 play significant role in *I. scapularis* attachment to host and facilitate blood-feeding. Finding of significant differences in XM_002400035 expression between LGTV-infected and uninfected- unfed/24 h-DF ticks, suggests that tick-borne flaviviruses do influence XM_002400035 in specific. Only XM_002400035, showed upregulation upon LGTV-infection and during blood-feeding, whereas in contrast, three other molecules (XM_002414778, XM_002434249 and XM_002400037) showed downregulation in LGTV-infected ticks (between unfed and 24 h-DF). Also, XM_002400035 showed high expression in post-fed ticks suggesting its multi-functional role. Reports show that LGTV-infection in tick cells modulates expression of specific organic anion transporting polypeptides that play essential functions in tick-pathogen interactions (Taank et al, 2018).

Furthermore, we repeatedly noted that LGTV promptly infect tick cells with higher viral loads detected at 72 h post-infection (Zhou et al, 2018). We consistently noted significant increase ($P < 0.05$) in LGTV-loads (at 72 h p.i.) from tick cell-derived exosomes when compared to cellular loads suggesting exosomes as

carriers of tick-borne LGTV. We previously showed the presence of full-length DENV2 RNA genome in mosquito cell-derived exosomes (Vora et al, 2018). We believe that infectious exosomes with full-length flaviviral RNA genomes are sufficient to bring in persistent infection in naive recipient cells. Current study provides clear evidence for enhanced exosomal production and release from LGTV-infected tick cells. These data correlated with higher exosome numbers counted from cryo-EM micrographs (Zhou et al, 2018). It has been shown that exosomes consist of lipid bilayer-enclosed spheres that participate in many cellular processes and serve as cargo for proteins, RNA/DNA/miRNA from donor to recipient cells (Sultana and Neelakanta, 2020). It is intriguing to note that XM_002400035 gene expression is upregulated in LGTV-infected tick cells and in exosomes. Expression of other six GRP molecules neither showed any changes in tick cells nor were detected in exosomes. Larvae and nymphs showed higher transcripts of XM_002400035 in comparison to adult ticks suggesting it is developmentally regulated. No differences were found for XM_002400035 transcripts between male/female ticks. In addition, XM_002400035 was induced in LGTV-infected fed larval ticks, molted unfed nymphs, and nymphs collected from 48 h-DF/48 h-PF repleted ticks during pathogen acquisition from infected mice into ticks. We noted upregulation of XM_002400035 in ticks during transmission of LGTV-loads to mice. Expression analysis and exosomal presence confirmed that other than blood-feeding, XM_002400035 facilitates LGTV survival, acquisition into ticks and transmission from infected ticks to naive vertebrate host via exosomes.

Studies have used silencing approach to disrupt function of several tick genes (Grabowski et al, 2019; Karim et al, 2011). Reduced LGTV-loads (both PrM transcripts and viral NS1-protein) in tick cells and in exosomes, upon XM_002400035 silencing provide strong evidence on its role in tick-LGTV-interactions. Silencing of XM_002400035 in LGTV-infected ticks severely affects blood-feeding and pathogen transmission. Tick body sizes/weights were drastically reduced in XM_002400035-dsRNA-treated ticks in comparison to respective mock-dsRNA-treated controls. These data correlated with significantly decreased LGTV-loads in XM_002400035-dsRNA-injected ticks. In correlation to reduced LGTV-loads, we also found that XM_002400035-silenced ticks were defective to transmit LGTV to mice as revealed by reduced pathogen loads in blood and other tissues. Overall, these data suggest an important role for exosomal GRP in pathogen acquisition, transmission and in tick blood-feeding. Detailed mechanism of how XM_002400035 facilitates virus survival in ticks and how this exosomal GRP aids in LGTV-transmission needs further studies.

During-feeding, ejected saliva suppresses blood clotting, complement activation, and modulates/inhibits host immune defense mechanism(s) that encounters arthropod-mediated modulations (Kazimírová and Stibraniova, 2013; Kotál et al, 2015; Nuttall, 2023). Tick exosomes secreted in saliva play key roles in delaying cell migration, repair, and wound closure, thus indicating their important function in easing blood-feeding and pathogen transmission at bite site (Neelakanta and Sultana, 2022; Sultana and Neelakanta, 2020; Wikel, 2018; Zhou et al, 2020). Pro-inflammatory-mediators CXCL-12 and IL-8 are abundantly expressed in mammalian skin and plays essential function during tissue repair and wound-healing processes (Zhou et al, 2020). Our

previous study showed that tick saliva, salivary gland/tick cell-derived exosomes modulates human chemokines, IL-8 and CXCL-12 (Zhou et al, 2020). However, tick exosomal molecule involved in this modulation is our future perspective.

Anti-vector vaccine approaches are proposed to block/reduce pathogen transmission from ticks to vertebrate host (Mahesh et al, 2023). Ticks uptake host proteins in blood meals while feeding (Mahesh et al, 2023). It is realistic that ticks could also ingest antibodies during-feeding on GST/GST-GRP immunized mice. Intriguingly, we noted that both fed nymphs and molted adult ticks that were infested on GST-GRP immunized mice had ingested the antibodies and hence showed higher binding to purified GST-GRP. The uptake of antibodies from murine host into tick bodies is intriguing, and our future studies will address the determination of these antibody quality, titers and longevity in mice and particularly in ticks. Lower viral loads in fed nymphs and molted adult ticks fed on GST-GRP immunized mice suggest that inhibiting exosomal GRP could be an ideal strategy to block pathogen transmission. Our results confirm that ticks ingest GST and GST-GRP antibodies via blood meal. Significant decrease in viral loads of ticks fed on GST-GRP immunized mice further suggest that targeting this exosomal molecule could be a best therapeutic approach to combat vector-borne diseases. Reduced viral loads in GST-GRP immunized mice demonstrate that blocking exosomal GRP affects transmission from infected ticks to vertebrate host. It is reported that immunization of guinea pigs with cement extract collected from adult female *I. scapularis* ticks resulted in partial protection of hosts (Lynn et al, 2022). Hematoxylin and eosin (H & E) staining showed increased inflammation in mice skin that allowed feeding of mock-dsRNA ticks when compared to mice skin that allowed feeding of XM_002400035-dsRNA ticks. We noted that GST-GRP immunized mice serum treatment reduced XM_002400035 transcripts and LGTV-loads in both tick cells and exosomes thus correlating to reduced exosome release. Exogenous treatment of GST-GRP delayed wound closures. Our study also revealed that GRP-antisera could potentially block the induced delay in wound healing, closure and repair process in HaCaT cells and in human keratinocytes primary cultures from the normal adult skin cells. The data with human keratinocytes suggest that antibodies against tick exosomal GRPs can be immediately tested on human skin as phase level studies in animals and perhaps on humans. Treatment of HaCaT cells with human GST-CXCL-12-protein showed enhanced wound closure and repair process. We noted restored levels of CXCL-12 transcripts in GST-GRP immunized mice skins that further supported wound-healing and repair process. A model is proposed to summarize all these current findings (Fig. 9G). Our studies with GST-CXCL-12-protein further strengthen it as a potential therapeutic. Active immunization studies with exosomal GRP impairs LGTV-transmission from ticks to mice. These results show that increased expression and secretion of biologically active GRP via salivary exosomes is important to promote the damaging effects from tick exosomes/saliva that impairs wound-healing and repair process. In turn, efficient production, and secretion of CXCL-12-protein at vector-host skin interface can control the devastating effects from tick salivary exosomes and exosomal GRP.

Collectively, our study highlights the role of an important exosomal GRP in tick blood-feeding, pathogen survival in ticks, pathogen acquisition from infected vertebrate host to tick, and in pathogen transmission to vertebrate host from infected ticks. In addition, we provide evidence that exosomal GRP has immuno-modulatory function to modulate human chemokine CXCL-12 signaling and is a potential candidate for development of transmission blocking vaccine. This study opens thoughts on how arthropod exosomes can mediate transmission of flaviviruses but also suggest a potential role of exosomal GRP for development of approachable strategies to prevent/control tick-borne pathogens and to combat vector-borne diseases transmitted by ticks and other medically important vectors.

# Methods

**Reagents and tools table**

| Reagent/Resource | Reference or Source | Identifier or Catalog Number |
|---|---|---|
| **Experimental models** | | |
| C57BL/6 J wild type mice (females) *Mus musculus* | Jackson Laboratory | # 000664 |
| C57BL/6 J wild type mice (females) *M. musculus* | Charles River Laboratory | # 027 |
| *Ixodes scapularis* larval ticks | BEI resources | # NR-44115 |
| *I. scapularis* nymphal ticks | BEI resources | # NR-44116 |
| *I. scapularis* adult ticks | BEI resources | # NR-42510 |
| *I. scapularis* (ISE6) cell line | BEI resources | # NR-12234 |
| Tick cell line ISE6 | ATCC | # CRL-11974 |
| Langat Virus (LGTV) | BEI resources | # NR-51658 |
| HaCaT cell line | Addexbio Technologies or Fisher Scientific | # NC0309203 |
| Primary cultures of Epidermal Keratinocytes | American Type Culture Collection (ATCC) | # PCS-200-011 |
| Vero cell line | ATCC | # CCL-81 |
| *Escherichia coli* cells (BL-21) | Fisher Scientific | # 69-449-3 |
| Tick cell-derived exosomes | In this study | |
| **Recombinant DNA** | | |
| pGEX-6P-2 | Millipore Sigma | GE28-9546-50 |
| pL4440 double T7 Script II vector | Fikrig Lab | |
| Purified CXCL-12-glutathione S-transferase (GST)-tagged protein | Santa Cruz Biotechnology, Inc | # sc-4654 |
| GST protein | In this study | |
| GST-GRP protein | In this study | |
| **Antibodies** | | |
| Langat virus NS1 antibody (clone 6E11) | BEI Resources | # NR-40308 |
| HRP-conjugated secondary Antibodies (goat antimouse/Rabbit IgG | Boster Biological Technologies | # BA1050 (mouse) # BA1054 (Rabbit) |

| Reagent/Resource | Reference or Source | Identifier or Catalog Number |
|---|---|---|
| Anti-beta-actin | Cell Signaling Technologies | #4970 |
| GST protein antisera | In this study | |
| GST-GRP protein antisera | In this study | |
| **Oligonucleotides and other sequence-based reagents** | | |
| Oligos for GRP transcripts | Designed in this study (provided in the Supplemental Table S2) and synthesized at IDT technologies | Integrated DNA Technology |
| Oligoes for dsRNA synthesis | Designed in this study (provided in the Supplemental Table S2) and synthesized at IDT technologies | Integrated DNA Technology |
| **Chemicals, Enzymes and other reagents** | | |
| Ampicillin | Sigma | #A0166-5G |
| BamH1 enzyme | Fisher Scientific | # FERFD0054 |
| Xho I enzyme | Fisher Scientific | # FERFD0694 |
| Kpnl enzyme | Fisher Scientific | # FERFD0524 |
| BglII enzyme | Fisher Scientific | # FERFD0084 |
| EcoR1 enzyme | Fisher Scientific | # FERFD0275 |
| DMEM media | Gibco #11320-033 | # 11320-033 |
| Dermal cell basal medium | ATCC | # PCS-200-030 |
| Keratinocyte growth kit | ATCC | # PCS-200-040 |
| L-Glutamine | Gibco | # 25030-081 |
| Penicillin–streptomycin solution | Gibco/Fisher Scientific | # 15140122 |
| Lipofectamine LTX reagent | Invitrogen/Fisher Scientific | # 11668500 |
| OptiMem | Gibco | # 31985070 |
| D-PBS | ATCC | # 30-2200 |
| Phosphate Buffered Saline (10X PBS), pH 7.2 | Sigma | # 806552 |
| Pierce BCA Assay Kit | Thermo/Fisher Scientific | # 23227 |
| Pierce™ protease inhibitor mini tablet, EDTA-free | Thermo/Fisher Scientific | # A32955 |
| SDS-Page gels | In this study (laboratory casted 12% gels) | |
| Agarose (BioReagent, Molecular Biology, low EEO) | Sigma | A9539-100G |
| Clarity Western ECL Substrate | Bio-Rad | #1705061 |
| Leibovitz's L-15 Medium, powder | Gibco | # 41300-039 |
| Tryptose phosphate broth | MP Biomedicals | # ICN1682149 |
| Bovine Cholesterol Lipoprotein Concentrate | MP Biomedicals | # ICN19147625 |

| Reagent/Resource | Reference or Source | Identifier or Catalog Number |
|---|---|---|
| Cytiva Grade 3MM Chr Filter Paper for Blotting | Fisher Scientific | 09-928-197 |
| Nitrocellulose membrane | Amersham GE | # 10600002 |
| Aurum total RNA mini kit | Bio-Rad | # 732-6820 |
| FBS | VWR | # 89510-186 |
| Multi-well plates | VWR | |
| dsRNA synthesis MEGAscript RNAi kit | Invitrogen/Ambion// Fisher Scientific | # AM1626 |
| iScript cDNA synthesis kit | Bio-Rad | # 1708891 |
| iTaq Universal SYBR Green Supermix | Bio-Rad | # L001752 B |
| 2X Universal SYBR Green fast qPCR Mix | ABclonal | # RM21203 |
| DNA Gel extraction kit | Qiagen | # 28704 |
| Pierce Hook GST protein Spin purification kit | G-Biosciences/ ThermoScientific | 16106 |
| TCE (2,2,2-Trichloroethanol, 99%) | Fisher Scientific | AAL0816314 |
| IPTG | Invitrogen/Fisher Scientific | 15-529-019 |
| Clarity Western ECL Substrate | Bio-Rad | # 1705061 |
| DMSO | Sigma | # D5879-500ML |
| Exo-Free FBS | System Biosciences Inc | EXO-FBSHI-50A-1 (50 ml) |
| Bovine serum albumin powder | Sigma | A9418-50G |
| Tween-20 | American Bioanalytical | # AB02038-00500 |
| Dry Milk powder | Fisher Scientific | 50-488-785 |
| RIPA lysis buffer | BioSciences | # 786-490 |
| EDTA-free protease inhibitor cocktail | Roche | # 11836170001 |
| Laemmli sample buffer | Bio-Rad | # 161-0737 |
| Coomassie Blue R-250 staining Solution | Bio-Rad | # 1610400 |
| DNA marker | Fisher Scientific | # SM0333 |
| 30% Acrylamide | Bio-Rad | # 1610156 |
| SDS | Sigma | #L3771-500G |
| APS (Ammonium Per Sulfate) | Columbus Chemical Industries | # 053500 |
| Tris HCl | American Bioanalytical | # AB02005-01000 |
| Tris base | Sigma | #T6066-1 KG |
| Glycine | Sigma | #G8898-1 KG |
| Hydrochloric acid (HCl) | Macron Fine Chemicals | #H613-46 |
| Acetate Buffer | ThermoScientific | #J60964.AK |
| Protein Plus ladder | Bio-Rad | # 1610373 |
| Complete Freund's adjuvant | Millipore Sigma | # 344289 |
| Incomplete Freund's adjuvant | Millipore Sigma | # 344291 |

| Reagent/Resource | Reference or Source | Identifier or Catalog Number |
|---|---|---|
| **Software** | | |
| ImageLab software | Bio-Rad | Part of CFXOPUS QPCR instrument |
| ImageJ2 software | NIH | https://imagej.net/ij/ |
| DNASTAR Lasergene | DNA STAR | https://www.dnastar.com |
| GraphPad Prism 6 software | GraphPad | Version 6 |
| Microsoft Office | Institutional (UT Provided Free Package) | |
| **Other** | | |
| Tick environmental Chamber | Parameter | Model 7311-N110-1C00000 |
| Ultracentrifugation unit | Beckman Coulter | 46910 |
| MRPS nCS1 nanoparticle analyzer | Spectradyne LLC | https://nanoparticleanalyzer.com |
| CFX-Opus QRT-PCR system | Bio-Rad | 12011319 |
| Cytation 7 imaging system | BioTek/Agilent | BTCYT7UMW |
| EVOS *M7000* auto fluorescence system | Thermo/Fisher Scientific | AMF7000 |
| Eppendorf FemtoJet microinjection system | Zeiss/Eppendorf | www.eppendorfna.com |
| Slide-A-Lyzer dialysis cassettes | ThermoFisher Scientific | # 87730 |
| Micro analytical balance | Sartorius | MCA224S-2S00-U |

## Study approval

All animal experiments in this study were performed with C57BL/6 wild-type mice (6-week-old females, obtained from Charles River (Catalog number 027)/Jackson Laboratories (C57BL/6 J, catalog number 000664, USA). All experiments were performed in strict accordance with recommendations in the Guide for care and use of Laboratory Animals of NIH, USA. Mice studies were performed based on protocol (# 2805-0321) approved by Institutional Animal Care and Use Committee (IACUC) and as previously reported (Ahmed et al, 2022).

## Sex as a biological variable

In this study, we used 6-week-old female mice from Charles River Laboratories or Jackson Laboratories. Female mice were considered, and sex is not a biological variable in this study.

## Mice, ticks, infections, and tick feeding on mice

Animal husbandry and administration of tranquilizer during animal experiments was performed as described (Ahmed et al, 2022). We also confirm that this study is performed in accordance with ARRIVE guidelines. Both male and female mice were tested first and since sex is not a biological variance in this study, we used 5–6-week-old female mice. *Ixodes scapularis* (larvae, nymphs, and adult ticks, Catalog numbers NR-44115 (Larvae) or NR-44116 (Nymphs) or adults NR-42510) were obtained from BEI resources/Center for Disease Control and Prevention (CDC). In our laboratory, tick rearing and maintenance was carried out in an incubator with 94% relative humidity under a photoperiod regiment of 14/10 h light/dark cycle at $23 \pm 2\,°C$. To examine the expression of cement-like glycine-rich transcripts in developmental stages of *I. scapularis*, we used unfed larvae, nymphs, and adults (male or female) ticks. Synchronous Langat virus (LGTV; LGT-TP21 strain, Catalog number NR-51658, obtained from BEI resources) infection in uninfected ticks was performed following the previous publication (Mitzel et al, 2007). A total of 64 unfed nymphs used in synchronous experiment were divided into two groups as 32 nymphs (16 in one tube), that were maintained as synchronously infected with LGTV. The other 32 ticks (16 in one tube) were kept as uninfected controls. Briefly, $1 \times 1e7$ pfu/ml of LGTV viral stock was mixed with 0.5 ml of complete L-15B300 medium to immerse one set of 32 nymphs at 34 °C for 45 min. Nymphs from uninfected control group were also immersed into L-15B300 without LGTV. To properly redistribute the medium over ticks, vials were vortexed after every 10 min. After bathing, ticks were washed with 1x phosphate-buffered saline (PBS) for five consecutive times. After washing, ticks were dried on a Whatman paper. Ticks (LGTV-infected or uninfected nymphs) were incubated in separate vials. Ticks were incubated in an environmental chamber for 17 days with relative humidity of 94%. Synchronously generated LGTV-infected unfed ticks were allowed to partially feed (for 24 h during feeding- DF) on C57BL/6 mice (6-week-old females from Charles River Laboratories, USA), During feeding, ticks were pulled off with forceps (and these are referred as 24 h DF ticks in this study). Uninfected ticks, to be used as controls, were fed on naive C57BL/6 mice.

Recently, utilizing LGTV as a model pathogen, we have established a method to infect naive *I. scapularis* nymphal/larval ticks by feeding these ticks on LGTV-infected murine host (Ahmed et al, 2022). In addition, we have also provided evidence for the transstadial transmission of LGTV from fed larval ticks to molted nymphs (Ahmed et al, 2022). Briefly, naive, or uninfected nymphal/larval ticks were fed on uninfected or LGTV-infected mice (injected intraperitoneal with 50,000 pfu/mouse) to demonstrate viral acquisition from murine host into the tick body. Larval ticks that acquired LGTV during feeding on infected murine hosts (acquisition experiment) were collected as fully fed larvae and were maintained in environmental chamber to molt into nymphal ticks. LGTV replicates and disseminates throughout the tick body during this incubation or molting process. These molted nymphs are used as unfed ticks. Nymphal ticks feeding on naive or infected mice were collected at two given timepoints of 48 h during feeding (DF) or 120 h post-feeding (PF) for further analysis. Fed larval ticks (referred as 120 h PF) were collected after full engorgement or repletion from uninfected or LGTV-infected mice and were further processed for molecular analysis. RNA/protein was extracted from these collected ticks and processed for testing LGTV loads as described (Ahmed et al, 2022).

## Cell culture, maintenance, and infection of in vitro cell lines

*I. scapularis* (ISE6 tick cell line; Catalog number NR-12234, obtained from BEI resources), human skin keratinocytes (HaCaT in vitro cell line, Catalog number NC0309203, obtained from Addexbio Technologies/Fisher Scientific, USA) and primary epidermal keratinocytes (normal, human, adult, Catalog number PCS-200-011, obtained from American Type Culture Collection (ATCC)), respectively, were used in this study. Wild-type LGTV was propagated in Vero cells (Catalog number CCL-81, obtained from ATCC) and cell culture supernatants from these cells were stocked as laboratory viral stocks. ISE6 cells were grown and maintained according to the instructions from Dr. Ulrike Munderloh (University of Minnesota, St. Paul, USA) and the culture methods are described in following publications (Dahmani et al, 2020; Namjoshi et al, 2023; Oliver et al, 2015; Zhou et al, 2018). Briefly, HaCaT cells were grown in complete DMEM medium supplemented with 10% heat-inactivated FBS. To perform infection kinetics, 5 × 1e5 ISE6 tick cells or were seeded in a 12-well plate, and infected with LGTV (multiplication of infection, MOI 1 for tick cells). Tick cells were collected at different timepoints (of 24 h, and 72 h post infection, p.i.) and processed for RNA isolation/protein extractions. Detailed information is elaborated in figure legends that corresponds to the data shown in different figures.

## Exosome isolation and quantifications

Exosomes were isolated and purified from tick cell culture supernatants by differential ultracentrifugation method and as described in our recent publications (Rajendran et al, 2021; Regmi et al, 2020; Vora et al, 2018; Zhou et al, 2020; Zhou et al, 2018). Schematic representation of exosome isolation is shown in detail in our previous studies (Vora et al, 2018; Zhou et al, 2018). Concentrated cell culture supernatants were used for exosome isolation. Exosome pellets were resuspended in PBS and stored frozen at −80 °C until further analysis. For reinfection experiments on naive recipient cells, freshly isolated exosomes were used. We implemented MRPS nCS1 nanoparticle analyzer (Spectradyne LLC, USA) on samples collected from uninfected or LGTV-infected or with mock or XM_002400035-dsRNA-treated tick cell-derived exosomes to examine the concentration and size of particles. Briefly, 5 × 1e5 tick cells were plated, after dsRNA transfections (for 48 h post transfection) followed by LGTV infection (for 48 h post infection). For treatment with serum (generated from GST or GST-GRP immunized mice), 5 × 1e5 tick cells were plated, and incubated (for 24 h) with serum was collected from immunized mice (sera 1:200 dilution) and then followed with LGTV infection (with MOI 1 for 72 h p.i.). Cell culture supernatant were collected and processed for exosome isolation using differential ultracentrifugation method. Samples were diluted 2000-fold, in PBST buffer, filtered through 0.22 μm membrane cartridges (Sartorius, Germany), and according to the manufacturer's instructions. Approximately, 5 μl of the diluted sample was loaded onto Spectradyne company's standardized cartridges/filters TS-300/c-300 (~50–300 nm) or TS-400/c-400 (~65–400 nm) to determine the particle size measurement range. These cartridges or filters are obtained from Spectradyne company and are specifically used for their nCS1 instrument. User-defined filtering parameters were applied to exclude false positive signals in all tested samples.

## RNA extractions, cDNA synthesis, and QRT-PCR analysis

Total RNA from ISE6 tick cells or exosomes-derived from ISE6 tick cells, uninfected and LGTV infected nymphs or larval ticks, or *I. scapularis* larvae, nymphs, adult (male or female), or mice tissue samples (such as blood, brain, liver, spleen, and skin) was extracted using Aurum total RNA mini kit (Bio-Rad) and following the manufacturer's recommendations. The iScript cDNA synthesis kit (Bio-Rad) was used to prepare cDNA from isolated RNA (1 μg) and used as template for the quantification of GRP transcripts, LGTV burden or actin transcripts that were determined with specific primers as shown in Appendix Table S1. QRT-PCR analysis was performed using iQ-SYBR Green Supermix (Bio-Rad, USA) or SYBR green mix (Abclonal, USA) and CFX96 or CFX-Opus QRT-PCR system (Bio-Rad, USA). Published specific primers were used to detect LGTV-RNA (Regmi et al, 2020; Zhou et al, 2018), mouse beta actin or tick beta-actin (Taank et al, 2018; Zhou et al, 2018). All oligonucleotides (forward or reverse primers) used for gene amplifications, dsRNA synthesis for exosomal XM-00240035, silencing efficiency or gene expression in this study are enlisted in Appendix Table S2 (see below). The standard curve for each gene fragment was generated using ten-fold serial dilutions starting from 1 to 0.000001 ng/μl of known quantities of respective fragments. Untreated samples served as internal controls.

## Immunoblotting analysis

Western blotting was performed as previously described (Vora et al, 2018; Zhou et al, 2020; Zhou et al, 2018). Briefly, 5 × 1e5 tick cells were seeded in six-well plates for overnight incubations. After transfections, followed by infections or infections alone total protein lysates were collected from either whole cell lysates or exosomes-derived from these cells and resuspended in modified RIPA buffer. Bradford assay (BCA kit from Pierce/Thermo/Fisher Scientific Inc.) was performed to determine the total protein amounts from cell or exosomal lysates. Mock or XM_002400035-dsRNA ticks (from transmission experiment) were collected after repletion. Five fully fed ticks were homogenized in RIPA-modified buffer, centrifuged and supernatant were processed for immunoblotting analysis. Total lysates were prepared from tick cells (25 μg), or exosomal lysates (10 μg from tick cells) or LGTV-infected mock-dsRNA/XM_002400035-dsRNA-treated fed nymphal ticks (8 μg) or from GST/GST-GRP (8 μg) or adult tick salivary glands (SG, 15 μg) or SG-derived EVs (10 μg) or from LGTV-infected tick cell (25 μg) or LGTV-infected mock-dsRNA-XM_002400035-dsRNA-treated fed nymphal ticks (10 μg). All proteins were separated onto 12% SDS-PAGE gels (reducing/non-reducing conditions). Total protein profiles served as loading controls. All total protein profile images are stained free gel images that were pre-casted in our laboratory using TCE (2,2,2-Trichloroethanol) that is incorporated into the SDS-PAGE gels during casting. TCE allows for the visualization and detection of even sensitive proteins through the UV-induced fluorescence. BSA or 5% milk buffer was used for 1 or 4–6 h/overnight blocking and probed with mouse monoclonal antibodies against Langat virus NS1 (clone 6E11, catalog number, NR-40308 from BEI Resources) with dilutions of 1:1000, respectively. For

immunoblots with GRP protein immunized mice serum, we used 1:100 dilution of serum for GST/GST-GRP and adult SGs/SG-EVs blot (reducing condition, 5% milk blocking buffer and all incubations performed at RT) or with 1:200 dilution for LGTV-infected mock-dsRNA/XM_002400035-dsRNA-treated tick cells or fed nymphal ticks total lysates blot (non-reducing condition, 5% BSA blocking buffer and all incubations performed at 37 °C). We performed both these immunoblots with anti-serum generated against XM_002400035 protein. We noted an intense band size around ~32 kDa in both adult tick salivary glands and SG-derived EVs total lysates. To confirm, if this is the band for XM_002400035 protein, we performed immunoblots with LGTV-infected mock-dsRNA and XM_002400035-dsRNA-treated tick cells and fed nymphal tick total lysates. As expected, we noted reduction in the ~32 kDa protein in XM_002400035-dsRNA-treated silenced samples confirming that this noted size band corresponds to XM_002400035 protein. The increased in XM_002400035 protein size could be due to several post-translational modification sites shown in Appendix Fig. S3. For horseradish peroxidase (HRP) conjugated (at dilutions of 1:5000) secondary antibodies (Goat Anti-Mouse/Rabbit IgG, HRP Conjugate, catalog numbers BA1050/BA1054, respectively) were obtained from Boster Biological Technology Co, LTD). Clarity Western Bright ECL kit (Bio-Rad) was used to detect the antibody binding. Chemidoc MP imaging system was used for imaging, and blots were analyzed with ImageLab software (Bio-Rad), and according to the manufacturer's recommendations. Densitometry analysis displaying band intensity for all immunoblots represented in this entire study are shown in Table S3 (see below).

## Wound healing/scratch assay and phase contrast microscopy

At bite site during ingestion of blood meal, *I. scapularis* ticks secrete different type of bioactive salivary factors in their saliva to evade host immune responses and immunological surveillance. Wound healing or cell scratch assays were performed to understand the functional role of exosomal GRP at the tick-host skin interface. Briefly, $5 \times 1e5$ ISE6 tick cells were plated, and exosomes were isolated from uninfected or LGTV-infected tick cells treated for 12 h with GST protein alone or GST-GRP or purified CXCL-12-glutathione S-transferase (GST)-tagged protein (as 2 μg per well/per group, for each protein; GST, GST-GRP, or CXCL-12) at 37 °C, followed by incubations on HaCaT cells. Note that CXCL-12 or SDF-1a (hBA-68) purified protein is produced from *Escherichia coli* bacterial lysates (>98%) and supplied as 35-kDa biologically active, GST-tagged fusion protein corresponding to 68 amino acids of the SDF-1a of human origin (Catalog number sc-4654, obtained from Santa Cruz Biotechnologies Inc., USA). For cell scratch assay, HaCaT cells (1–3 × 1e6) or human keratinocyte primary cultures (5 × 1e5) were seeded in six/twelve-well plates with Dulbecco's modified Eagle's medium (DMEM) containing 5% heat-inactivated FBS or with dermal cell basal medium supplemented with keratinocyte growth kit components (obtained from ATCC, USA). After overnight incubation, culture medium was changed, and fresh medium supplemented with 10% FBS was given to the HaCaT monolayer of confluent cells or complete dermal cell basal medium with keratinocyte growth kit components was given to the human keratinocytes. Images of HaCaT cell monolayers or human

keratinocytes were captured as before treatments or at different timepoints of 0, 4, 16, 20, 24, 48, and 72 h post treatments with respective groups. The group of images collected before treatment or scratch are shown as before scratch. Monolayers of HaCaT confluent cells or human keratinocytes cultures were scratched (in the middle of the well, to generate a wound) with a sterile 10 μl pipette tip. Immediately after scratch, monolayers of HaCaT cells were treated with tick cell-derived exosomes (20 μl of each respective group) or both HaCaT cells and human keratinocytes were either treated with GST-GRP protein alone (2 μg/ml), or with GST-GRP antisera (from GST-GRP immunized mice) as 1:100 dilution. HaCaT cells or human keratinocytes were observed, and numerous phase contrast/bright field images were collected (using EVOS M7000 auto fluorescence system, at 10× magnification, or imaged using Cytation 7 imaging system with 4× magnification and with a scale bar of 275 or 1000 μm) at different time points. HaCaT monolayers/human keratinocytes with scratches but without treatments were considered as untreated (UT) internal controls in each wound healing assay. ImageJ2 software (NIH, Bethesda, MD, USA) was used to determine cell migration under the aforementioned culture conditions. Wound diameters were calculated as a percentage of distance of cells x (h) post-scratching (distance of cells at scratching time point). For details on this wound diameter measurements, please refer our previous study (Zhou et al, 2020).

## Bioinformatics alignments and prediction analysis

After combing the *I. scapularis* genome, tick cement-like glycine-rich amino acid sequences were obtained from GenBank and analyzed in ClustalW program in DNASTAR Lasergene, to align the collected amino acid sequences (with other orthologs), and as described earlier (Regmi et al, 2020; Taank et al, 2018). The phylogenetic tree of various GRPs sequences was generated using the Neighborhood Joining (NJ) method with DNASTAR Lasergene to reveal evolutionary relationships. The conserved motif/domain sequence is shaded for matching residues in black color for easy identification. The percent identity and divergence percentages were obtained from prediction analysis. TMHMM server v.2.0 (https://services.healthtech.dtu.dk/service.php?TMHMM-2.0) was used to predict transmembrane and inside or outside regions of GRPs, and as described in our previous study (Taank et al, 2018). N-myristoylation, casein kinase II phosphorylation, N-glycosylation and protein kinase C (PKC) phosphorylation sites in GRPs sequences were individually analyzed and predicted at PROSITE (http://prosite.expasy.org/) and as described (Sultana et al, 2015; Sultana et al, 2016). The SignalP 5.0 server (https://services.healthtech.dtu.dk/service.php?SignalP-5.0) was used to predict the presence of signal peptides and the location of their cleavage sites in GRPs.

## dsRNA synthesis, tick cell transfections, tick microinjections and acquisition/transmission studies

The dsRNA synthesis was carried out following standard procedures (Khanal et al, 2022; Neelakanta et al, 2010; Ramasamy et al, 2020; Sultana et al, 2010; Taank et al, 2020; Vora et al, 2017). First, we amplified a 136 bp fragment of XM_002400035 from *I. scapularis* tick cDNA sample and cloned it into KpnI and BglII enzyme restriction sites of the L4440 double T7 Script II vector

(Appendix Fig. S7B). The fragment of interest was PCR amplified by using specific primers comprising the KpnI and BglII restriction enzyme sites as listed in Appendix Table S2. PCR amplified *I. scapularis* GRPs fragment was gel purified using gel extraction kit (Qiagen, USA) and following user's manual instructions. The amplified fragment is cloned into L4440 double T7 Script II vector in BglII-KpnI enzyme restriction sites. MEGAscript RNAi kit (Invitrogen/Ambion, Thermo/Fisher Scientific, USA) is used to synthesize the dsRNA complementary to *I. scapularis* XM-00240035 sequence. We generated dsRNA targeting *I. scapularis* exosomal XM_002400035 sequence as described in (6). Two subsequent elution of dsRNA are shown in the gel image (Appendix Fig. S7B). For dsRNA transfections and silencing of XM_002400035 molecule, we plated $5 \times 1e5$ ISE6 tick cells grown in complete L-15B300 media containing 5% regular FBS (exo-free), and cells were allowed to adhere overnight. Lipofectamine reagent (Invitrogen, Thermo/Fisher Scientific) was mixed with 750 ng of dsRNA for transfection of tick cells. Tick cells were transfected with either mock-dsRNA control (empty plasmid L4440) or XM_002400035-dsRNA. To recover cells from transfection stress, 2X L-15 recovery media with (10% FBS, exo free) was added to cells, after six hours of post transfection. Followed by recovery, LGTV (MOI 1) infection was performed after 24 h of dsRNA treatment. Numerous phase contrast images were collected (at 10× magnification, and with a scale bar of 200 µm) at different time points (of 4 h or 24 h post transfection (p.t.), or at 24 h or 48 h p.i.) from each group. Representative images captured using the EVOS auto-fluorescence System M7000 at 4 h post transfection (p.t.), or 24 h p.t., and 24 h post infection (p.i.) or 48 h p.i to check cytotoxicity and morphological changes are shown. Tick cells were collected at 48 h p.i. (or 72 h post transfection) for further analysis. QRT-PCR was performed to determine silencing efficiency of XM-00240035 transcripts by using specific primers as mentioned in Appendix Table S2. For microinjections in ticks, *I. scapularis* nymphs were microinjected with either mock or XM_002400035-dsRNAs (~4.2 nl/tick or ~10 ng/tick) using Eppendorf FemtoJet microinjection system and as described in our previous publications (Khanal et al, 2018; Ramasamy et al, 2020; Taank et al, 2020; Taank et al, 2018). Microinjected ticks were placed inside incubator for 24 h (for recovery) with settings of photoperiod time set at 14/10 h light and dark exposure, humidity at 90–94% and temperature at $23 \pm 2\,°C$. Uninfected nymphs, microinjected with XM-00240035-dsRNA or with mock control-dsRNA were later (after 24 h recovery) fed on either uninfected or LGTV-infected mice for acquisition studies, respectively. LGTV-infected nymphs microinjected with XM-00240035-dsRNA or with mock control-dsRNA were fed on uninfected mice for transmission studies. Fully engorged/repleted ticks were collected (at 48 h post feeding) and processed for RNA/protein extractions and were analyze for XM_002400035 expression (silencing efficiency) and LGTV loads by using QRT-PCR or immunoblotting analyses. All oligonucleotides are listed in Appendix Table S2.

Cloning of exosomal GRP, protein expression and purification. For cloning of exosomal XM_002400035 fragment, we chose all the information from GenBank. The predicted protein size of tick exosomal GRP is 62 amino acids (please refer to GenBank protein accession number EEC20123.1 at https://www.ncbi.nlm.nih.gov/protein/215510670). Also, please refer to the nucleotide accession number XM_002400035 for more details. We amplified a fragment

of (105 bp) from exosomal XM_002400035 sequence that contained *Bam*HI and *Xho*I restriction enzyme sites and cloned into expression vector pGEX-6P-2 with glutathione S-transferase (GST) tag (Bioexpress, Kaysville, UT, USA). Our GST-protein cloning system with pGEX-6P-2 plasmid comes with the PreScission cleavage site that allows the cleavage at low temps. The PCR amplified XM_002400035 fragment cloned into the vector, is a polypeptide or protein size of ~10 kDa, that is coupled to GST protein (26 kDa), thus leading to a protein product size of 36 kDa GST-GRP. The selected clone was confirmed by sequencing. *Escherichia coli* BL-21 competent cells were transformed with the ligated product and Ampicillin antibiotic (50 µg/ml) selective clones were picked on LB agar plates. Empty pGEX-6P-2 vector transformed into BL-21 served as GST control. We induced and expressed GST or GST-tagged GRP in *E. coli*. We next purified recombinant GRP conjugated with GST. The GST and GST-GRP was purified from BL-21 cells using a Hook GST protein Spin purification kit (G-Biosciences, St. Louis, MO, USA) and following manufacturer's suggestions with some modifications. Before harvesting by centrifugation, *E. coli* was induced with 1 mM IPTG at 37 °C (Appendix Fig. S10A). The recombinant GST-GRP (containing 10 mM glutathione) was eluted in 1x phosphate-buffered saline (PBS). Slide-A-Lyzer dialysis cassettes (10 k molecular weight cut-off, Thermo/Fisher Scientific, USA) was used to dialyze the purified protein in 1x PBS at 4 °C, overnight and by following the company's instructions. Protein estimations was performed with Pierce BCA protein assay kit (Thermo/Fisher Scientific, USA). Dialyzed proteins (GST or GST-GRP) were used for mice immunization studies.

## Mice immunization

Active immunization experiment was performed with C57BL/6 wild-type mice (6-week-old female mice obtained from Jackson Laboratories, USA). At day 0, 14, and 28 mice were actively immunized (intraperitoneal) with GST ($n = 5$), or GST-GRP purified protein ($n = 6$) (50 µg/mouse for the first dose with Complete Freund's adjuvant and 25 µg/mouse for second and third booster dose with Incomplete Freund's adjuvant). Freshly molted LGTV-infected nymphal ticks were allowed to feed on these immunized mice (after 14 days of third booster) and ticks were collected after repletion (Appendix Fig. S10B). A detailed schematic representation is shown for the immunization experiment. After euthanasia, the mouse skin samples were collected for ELISA assays and QRT-PCR analysis. After tick repletion and mice euthanasia, mice blood was collected through cardiac puncture to prepare serum from immunized mice. In addition, mice tissues were collected for further molecular analysis.

## Enzyme-linked immunosorbent assay (ELISA)

We confirmed the specificity of serum collected from GST or GST-GRP immunized mice with GST/GST-GRP recombinant proteins and mouse skin lysates. ELISA were also performed to show the presence of exosomal GRP antibodies in nymphs fully fed and repleted or adult ticks molted from those repleted nymphs fed on GST-alone or GST-GRP immunized mice. First, we pre-coated 96-well plates (Nunc, Thermo/Fisher Scientific, USA) for overnight at 4 °C with (a) 500 ng of GST or GST-GRP purified proteins, or (b)

with 250 ng of adult *I. scapularis* tick salivary glands (SGs) or EVs-derived from SGs or (c) LGTV-infected mock-dsRNA/XM-00240035-dsRNA-treated tick cells or d) with fed nymphal ticks lysates probed with GST or GST-GRP immunized mice serum. In addition, we coated purified proteins (GST or GST-GRP, each protein as 500 ng) probed with LGTV- infected tick lysates (500 ng) collected from repleted nymphal ticks or adult ticks that were molted nymphal ticks fed and repleted from GST-alone or GST-GRP immunized mice. In this ELISA assay, we coated the wells with purified GST or GST-GRP proteins as substrates. We probed the proteins with infected tick lysates from fed nymphs or molted adult ticks (collected from GST/GST-GRP-immunized mice) to bind the antibodies present in tick lysates. The binding was detected with anti-mouse HRP conjugated secondary antibodies. Empty wells served as reference controls in all ELISA assays. Blocking buffer (1% BSA in 1x PBS and 0.1% Tween-20) was used at 37 °C for 1 h. Serum collected from GST or GST-GRP immunized mice was diluted as 1:200 for ELISA with purified proteins (Appendix Fig. 7B) or with 1:100 (for GST/GST-GRP and adult SGs/SG-derived EVs lysates)/1:200 dilution (for lysates from LGTV-infected mock-dsRNA/XM-00240035-dsRNA-treated tick cells or fed nymphal tick lysates) (Appendix Fig. 11). Wells were washed three times with wash buffer (containing 0.05% Tween-20 and 1x PBS), and anti-mouse secondary antibody conjugated with horseradish peroxidase (Goat Anti-Mouse IgG Secondary Antibody, HRP Conjugate, catalog number BA1050 was obtained from Boster Biological Technology Co, LTD) was added at a dilution of 1:2000 and incubated for 1 h at 37 °C. Followed by three washes, 100 µl of peroxidase substrate (Sure Blue, KPL, USA) was added and incubated at 37 °C for 15 min. The reaction was stopped by adding 50 µl of stop solution and absorbance was measured at 450 nm in CYTATION 7 reader (BioTek, USA).

### Skin biopsy sample collections and histological procedures

Skin biopsies (of 2–4 mm) with attached mouthparts of ticks were collected from mice that were fed with mock/XM-00240035-dsRNA-treated ticks. Skin tissues were collected using a sterile dermal biopsy punch. Tissues from skin biopsies were fixed immediately in formalin (10% neutral buffered formalin), and samples were stored at 4 °C and then processed at Histology core facility in the College of Veterinary Medicine, University of Tennessee Knoxville, USA, following standard staining with hematoxylin-eosin (H&E) staining and examination by light microscopy. Inflammation was evaluated by the pathologist using the scale none, minimal, mild, moderate, marked, and severe. Samples were blindly evaluated with no information provided to the pathologist on which samples belonged to different treatment groups. A true grading scheme was not created due to low number of samples.

### Statistical analysis

GraphPad Prism 6 software was used to observe statistical significance in data sets. We have performed the one-way ANOVA with Tukey's test for multiple group comparison. In all other panels, we have used the non-parametric Mann–Whitney U test for statistical analysis. A table for Mann–Whitney U test (Appendix Table S4) has been provided in the Appendix information that shows the median for each sample. The main criteria/reasons for

the selection of different statistical test in this study are the number of variables, types of data/level of measurement (continuous, binary, categorical), the scientific question being answered, the data structure and the type of study design (paired or unpaired). Median values in experimental samples compared using Mann–Whitney statistical analysis are provided in Appendix Table S4. Error bars represent mean (+SD) values, and *P*-values of <0.05 were considered significant in all analyses and shown at relevant places. In each experiment data represents biological replicates. The exact number of samples (*n*) for each figure panel representing multiple experiments is included in the respective figure legends.

## Data availability

This study includes no data deposited in external repositories or other places. No human-subject data is included in this study.

The source data of this paper are collected in the following database record: biostudies:S-SCDT-10_1038-S44318-026-00709-z.

## Peer review information

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

## Acknowledgements

We sincerely acknowledge the useful resource of ticks from BEI resources/CDC. Following reagents were provided by Center for Disease Control and Prevention for distribution by BEI Resources, NIAID, and NIH: *Ixodes scapularis* Larvae (Live), NR-44115; *I. scapularis* Nymph (Live), NR-44116 and *I. scapularis* Adult (Live), NR-42510. We are extremely thankful to Dr. Michael L. Levin and others for ticks rearing and shipments. Following reagent was obtained through BEI Resources, NIAID, NIH: (a) Langat Virus, TP21, NR-51658. (b) Tick Cell Line ISE6 Derived from *Ixodes scapularis* Embryos, NR-12234. (c) The following reagent was obtained from Joel M. Dalrymple - Clarence J. Peters USAMRIID Antibody Collection through BEI Resources, NIAID, NIH: Monoclonal Anti-Langat Virus Nonstructural Protein 1 (NS1), Clone 6E11 (produced in vitro), NR-40308. This study was supported by funding from National Institute of Allergy and Infectious Diseases (NIAID)/National Institutes of Health (NIH) (Award numbers R01AI141790 and R01AI141790-05S1 to HS).

## Author contributions

**Waqas Ahmed**: Formal analysis; Validation; Investigation; Visualization; Methodology; Writing—review and editing. **Wenshuo Zhou**: Formal analysis; Validation; Investigation; Methodology; Writing—review and editing. **Md Bayzid**: Validation; Methodology; Writing—review and editing. **Denae Nadine LoBato**: Methodology; Writing—review and editing. **Kehinde D Fasae**: Visualization; Methodology. **Girish Neelakanta**: Resources; Software; Formal analysis; Supervision; Validation; Investigation; Visualization; Methodology; Writing—review and editing. **Hameeda Sultana**: Conceptualization; Resources; Data curation; Software; Formal analysis; Supervision; Funding acquisition; Validation; Investigation; Visualization; Methodology; Writing—original draft; Project administration; Writing—review and editing.

Source data underlying figure panels in this paper may have individual authorship assigned. Where available, figure panel/source data authorship is listed in the following database record: biostudies:S-SCDT-10_1038-S44318-026-00709-z.

## Disclosure and competing interests statement

The authors declare no competing interests.

