## [Peer Review File · The EMBO Journal]

Arthropod exosomal glycine-rich protein as a potential vaccine candidate

Waqas Ahmed, Wenshuo Zhou, Md Bayzid, Denae LoBato, Kehinde Fasae, Girish Neelakanta, and Hameeda Sultana

Corresponding author: Hameeda Sultana (hsultana@utk.edu)

Review Timeline:

Submission Date:	6th Jun 24
Editorial Decision:	27th Sep 24
Revision Received:	28th Jul 25
Editorial Decision:	17th Sep 25
Revision Received:	19th Dec 25
Editorial Decision:	9th Jan 26
Revision Received:	9th Jan 26
Accepted:	15th Jan 26

Editor: Yehu Moran

Transaction Report:

Dear Prof. Sultana,

Thank you for submitting your manuscript for consideration by the EMBO Journal. I would to start by apologizing again for the unusually long time it took to get a sufficient number of referee reports for your manuscript. Finally, it has now been seen by three referees whose comments are shown below.

Given the referees' positive recommendations, I would like to invite you to submit a revised version of the manuscript, addressing the comments of all three reviewers. I should add that it is EMBO Journal policy to allow only a single round of revision, and acceptance of your manuscript will therefore depend on the completeness of your responses in this revised version.

In light of our policy of a single round of major revision and in light of the fact that some of the concerns raised by the referees seem to be quite substantial, I would suggest that as an initial stage in your revision process you will prepare with the other authors a revision plan (major points and planned additional experiments and/or analyses, written in brief) and send it to me in advance. In our experience a revision plan approved by the editor can improve the acceptance chances in the next round.

Thank you for the opportunity to consider your work for publication. I look forward to your revision.

Yours sincerely,

Yehu Moran
Academic Editor
The EMBO Journal

- a Reagents and Tools Table as part of the Methods section, which can be downloaded from our author guidelines

We realize that it is difficult to revise to a specific deadline. In the interest of protecting the conceptual advance provided by the work, we recommend a revision within 3 months (26th Dec 2024). Please discuss the revision progress ahead of this time with the editor if you require more time to complete the revisions.

Referee #1:

The manuscript "Arthropod exosomal cement protein as a potential vaccine candidate effectively reduces tick blood-feeding and pathogen transmission" describes target validation of several *Ixodes scapularis* tick proteins in feeding and transmission of the Langat virus by ticks. Overall, the manuscript is written with clarity, and it is difficult to follow. The central objective of this study is not apparent and makes it difficult to understand the basis of this work. Significantly, weaving the word "cement" in this manuscript is not based on evidence or background data in literature. It is unclear why proteins in this study were referred to as cement proteins as the authors provide no evidence that the proteins in this manuscript were involved with tick cement formation. Notably, the publication (PMID: 36494396) that described proteins that form *Ixodes scapularis* cement was not cited in this study demonstrating that the review of literature on tick cement proteins was not well done. Furthermore, the authors mention throughout the manuscript that the proteins are heat shock proteins; however, XM_002400035 is indicated as a glycine rich protein, please verify your information.

Additional queries/suggestions to improve the manuscript are below.

Title. The title is wrong, there is no evidence that the proteins studied in this study are cement proteins, please revise.

Abstract. The abstract is too descriptive, authors could consider highlighting in the abstract major outcomes in this study.

Introduction. The objective for this study is unclear, please provide your reasoning for this study here. Literature review is incomplete,

Results. The results section is not well written. Given that the materials and methods are in supplemental, authors should provide a couple of sentences on methods used. For instance, on RNAi silencing, what does mock dsRNA mean? This is confusing to me. Did you inject non-tick related dsRNA, this is what is required for a control. Please specify in results. Provide quantitative data. For instance, this sentence "These data implicate that reduced body weights is likely a result of decrease blood-feeding by these cement-knockdown ticks" does not mean much, please state % of folds reduction. Similar descriptive statements must be accompanied by quantitative data.

Figures. Handwritten labels on tick images don't look good, please type.

Referee #2:

Tick-borne diseases are in expansion worldwide with serious risks for human and animal health. The need of vaccines is urgent and the concept of anti-tick vaccine is regularly mentioned as a solution to limit these diseases. Up to now, numerous tick saliva molecules have been identified but none have been shown to have a real protective effect to stop the transmission of pathogens.

The manuscript by Ahmed et al. describes the identification of a cement protein from tick saliva involved in the transmission of Langat virus. Among seven cement proteins deposited in genbank, the authors selected one particularly involved in the process of *Ixodes scapularis* blood feeding. The Langat virus specifically upregulates this protein in exosomes released during the tick bite and uses these exosomes to facilitate their transmission.

Using different techniques of RNAi silencing, they show that the virus interacts with this protein in tick cell lines and in mouse model. The silencing of the cement protein reduces the virus transmission. They propose this cement protein of tick saliva as a

potential vaccine candidate to reduce the tick blood meal but also to reduce the virus transmission. All the project is well structured with numerous experiments to validate their hypotheses, and thoroughly documented. They conducted different techniques in vitro and in vivo to demonstrate the major role of this cement protein.

Few comments on the study:

However, the concept of an anti-tick vaccine based on a tick saliva protein isolated from cement is not new. Labuda and Nuttall have already demonstrated the role of cement protein in the transmission of tick-borne encephalitis virus, and have proposed testing a specific cement protein as a vaccine to block the transmission of tick-borne encephalitis virus (Labuda et al. Plos Pathogens 2006 - cited by the author). More recently, Fikrig's group at Yale has also proposed testing the cement protein as a candidate vaccine (Lynn et al. TTBDs 2022). Numerous tick saliva proteins have already been identified in different ticks, but none has really shown any effect in blocking transmission of infectious agents. At most, the tick's blood meal is reduced (as shown here) and the female lays fewer eggs, or an allergic reaction is induced by immunization with the saliva protein, leading to rejection of the tick.

- I am surprised that none of the 7 proteins is clearly identified as a precise protein. Different studies characterized different tick cement proteins in different tick genera.
- Line 27-28 : "The molecular mechanism(s) for cement secretion at feeding pit and their role in tick-pathogen interactions is not clearly understood". I do not agree. The cement has been very well characterized and its function is clearly established.
- Figure 4. How do the authors explain that the cement protein they identified is not upregulated in male and particularly female ticks? The blood meal of female lasts for several days and cement is essential to anchor the ticks to the skin.
- viral infestation of larvae or nymphs: is immersing nymphs in a viral suspension really effective for infecting ticks? I have a few reservations. I've never really trusted this type of infestation, it's too artificial.
- I also have some reservations concerning the use of the HACAT cell line ; primary human keratinocytes available commercially would be more reliable.
- minor errors:
 - o line 483: Replace "Its shown" by "It shows"
 - o line 489: Replace "it as developmentally regulated" by "it is"
 - o line 551 : replace "our study highlights an important exosomal" by "our study highlights the role of an important exosomal..."
 - o Line 816: figure legend: "seven cement" genes or molecules ?
 - o Line 829: same as line 816

Referee #3:

EMBO J

EMBOJ-2024-118127

Ahmed et al

Arthropod exosomal cement protein as a potential vaccine candidate effectively reduces tick blood-feeding and pathogen transmission

This is an interesting, if very dense, manuscript that focuses on an *Ixodes scapularis* tick cement component, cement-XM_002400035 molecule, and its (exosomal) roles in tick bio/pathology (particularly in acquisition and transmission of LGTV), and whether this molecule could be a target for effective immunization. As tick-borne diseases are on the rise worldwide, it is imperative to determine mechanisms of disease transmission with the hopes of determining effective ways to prevent these diseases. As such, the studies are largely well done, but the manuscript is a difficult read. Much of this has to do with following all the life stages and feeding cycles of the ticks, but it begins with the "cement-XM_002400035 molecule" designation. While this is an official term, presumably, it becomes cumbersome as often as it is repeated in the text, particularly in the context of mock and dsRNA inhibition. Could there be a way to shorten it after the first time it is denoted?

For other areas, please be careful to note when you are measuring or describing transcripts vs protein, especially after you generate the murine antisera that can be used to identify or measure protein.

Also, there are many instances of need for articles ("a", "an", "the") and subject-verb agreement, particularly in singular or plural forms. I have identified some of those, but then lost further energy to do so. I will leave the rest to the authors and copy editors.

Comments follow in order of the manuscript.

Intro

Line 52: the *and* between tick species names probably does not need to be italicized.

Lines 54-55: it is unclear if "Langat Virus-LGTV," is the model pathogen similar to TBEV/POWV (it does not seem to be?), or

what is the model pathogen?

Line 63: *in* should be deleted.

Lines 78-79: you may wish to state that 64TRP as an antigenic vaccine component shows the ability to control tick-borne diseases.

Line 80: I think it would be clearer to remove *establish to* from the sentence.

Line 111: *chemokine* CXCL-12 should be singular.

Supp M&M etc

Line 54: perhaps should say "Uninfected ticks, to be used as controls, were fed on naïve C57BL/6 mice."

Line 83: *detailed* information....

Lines 105-106: need to mention that those filters are for the Spectradyne instrument, as this follows right after Sartorius membrane filtering.

Line 134: *supernatants* (plural)

Lines 158-159: there are two "obtained from" mentions; it appears it is only Boster Bio that is the source?

Line 170: *assays* were performed

Line 204: "...sites...*were* individually analyzed...."

Lines 285-292: this might be easier to read as a bulleted list; there is a lot going on, here.

Results

Bioinformatics; Supp Fig S3 - it might be helpful to highlight (say, in yellow, or bold font) EEC20123.1 in the axes locations, or direct the reader to it saying it is on the far left of each graph in Supp Fig S3

Results Section: LGTV-infection selectively upregulates expression of cement-XM_002400035 in tick cells/tick cell-derived exosomes and enhances exosome secretion

Line 180+: Regarding Fig 3 and Supp Fig S5, what is the justification for using actin message for normalization of cement component messages in exosomes? Would it not be better to use particle count or mass of exosomes, as it is not clear in the EV field that any particular protein or message is universally equally loaded into exosomes/EVs? Second, as mRNAs in exosomes are often fragments, is this full-length cement-XM_002400035 (or other cement messages)? Third, would it not be more important to look for the protein in exosomes instead of mRNA, ie, what is the justification for looking at the message rather than the protein product? I realize that protein detection may be difficult, and it is somewhat shown in Fig 7C (not the greatest blot), but in the biology of cement components in exosomes, what do the transcripts actually do? Finally, there needs to be some mention, perhaps in the legend, in Supp Fig S5 that the quantities (y-axes) are quite different, in some cases, several orders of magnitude (see Sig S5 E vs F, or even vs G). Thus, while XM_002434249 may not change significantly in cells upon infection, there is far more message under all conditions compared to XM_002400035.

For the exosome size measurements, it seems as if the average diameters are smaller for exosomes from infected cells, although this can be difficult to discern at a glance. Is it possible at that increased exosomes numbers from the infected cells could be due to release of LGTV itself, or is the virus too small to be detectable with the Spectradyne equipment?

Section: RNA interference of cement-XM_002400035 impacts its expression and LGTV-loads in tick cells/exosomes and in nymphal ticks during pathogen acquisition

Supp Fig S7: I think the word you mean is "elution", not "elusion".

Fig Legend 5: "Total protein profiles are stain-free gel images". Are those not actually *stained* gel images?

I have the same comments (from above) regarding the loss of viral replication or shedding as part of the reduction in exosome numbers (and loss of genetic and protein components seen in Fig 5).

For Supp Table S3, was actin ever accounted for in the densitometry?

Line 313 and Fig 6E; you may wish to lighten the blot for NS1 as it is not easy to see the differences between mock and dsRNA-treated ticks with this image.

Line 315: ..."this *immunoblot*..." [singular]

In the Fig 6 Legend, lines 954-955, should be "A group of repleted ticks *was* also immediately weighed...."

Again, as mentioned above, "stain-free" does not make sense, do you mean "destained"?

Fig 6 legend line 974, you do mean panels *6* F-H?

Line 323: Should say "We also observed reduced viral loads in mice tissues (brains and spleens) *from mice* that allowed feeding...." Otherwise, it sounds like the reduced viral loads allowed feeding of ticks.

Line 327: *treated*

Supplemental section, legend to Supp Fig S10, line 592, should just be *engorge and were collected*

Supp Fig S11, legend, lines 616-617, the terms "Mock-I" and "dsRNA-I" never show up in the figure.

Results Section: Immunization with exosomal cement protein impairs LGTV-transmission from infected ticks to naïve vertebrate host

Line 340: *bind*

Line 344: presumably these are "mice serum *antibodies*..."

Lines 371-375, while this is in the Supp Materials and Methods, the detection of anti-cement Abs in the nymphs/adults deserves a little better technical explanation here in the results, because this is rather fascinating. Here, GST or GST-cement proteins were used as capture substrates for Abs in the tick lysates, followed by anti-mouse Abs for detection. Given that you were using the mouse sera as a probing agent for most of these assays, this is a necessary clarification.

Figure 7 legend:

Line 991: *are* shown

Lines 991-993: This is a difficult sentence given the number of / in it and the number of immunoblot targets. Also, it should be "cement-XM_002400035-*dsRNA*."

Line 996: "A group of ticks *was* ..weighed..." (In this case you specifically stated a single group).

Lines 1008+: you begin referring to panels "10B" etc through here - I presume you mean panel 7.

Results Section: Silencing of exosomal cement reduced inflammation at tick bite site in mice and exogenous GST-CXCL-12 treatment closed wounds in human keratinocytes

There needs to be better explanation of Fig 8 and Supp Fig S12, in particular, the arrows/arrowheads are too small, and need have better distinguishing features (particularly in the denser cellular backgrounds). For the wound/scratch assay, there needs to be some explanation of the expected results vs what is seen (or in conjunction with what is seen), because there are multiple levels of complexity inherent in the experiment. Are there any statistics applied to these measurements? In Fig 8E, what is GST-C6?

Figure 8 legend

Lines 1037-1038: LGTV-infected and mock-dsRNA-treated (M) or LGTV-infected and cement-XM_002400035-dsRNA-treated groups *are* shown

We do not see "M" anywhere in the figure?

Line 1058: this is just a question - would Langerhans cells (ie, epidermal DCs) and dermal DCs have any impact on tick feeding?

These would seem more immediately responsive than circulating blood DCs. Just wondering.

Line 1059: "tick cement exaggerates residential keratinocytes..." to do what?

Discussion

Line 483: *It has been shown that*

Lines 485-86: Please be clear here, and in general, when you say "It is intriguing to note that only cement-XM_002400035 is upregulated in LGTV-infected tick cells and in exosomes" what you refer to - protein, or transcript? Since you have potential antisera now, it will not be inherently obvious when you mean transcript vs protein.

Paragraph starting at line 522: the uptake of antibodies from host to ticks is fascinating, but likely requires high titer of Ab. Is there anything known about how much Ab may be "retrotransmitted" to ticks by any given animal, much less an immunized one? That would be an important determinant in further pathogen transmission/inhibition outside the host. While beyond the scope of this already dense manuscript, it could also conceivably be quantified in future experiments where quantity of Ab transmission to ticks is compared to Ab titer in the host.

Lines 538-9: "Exogenous treatment of GST-cement-XM_002400035 delayed wound closures." A truly interesting experiment here would be to show that anti-cement antisera could alleviate the cement-induced wound closure delay. Do you have enough material remaining to do that?

One limitation must be stated in this study, and that is for experiments involving murine antisera and GST as a target, the mice from this study did indeed have Abs vs GST (as well as the cement Ag; Fig 7C). Thus, there could be background reactivity to the GST component. For future experiments, it might be suggested to use a cleavable GST linker to remove that following purification and prior to immunization.

There should also be some discussion as to the choice of cement fragment/polypeptide used for immunization, ie, the putative protein sequence should be shown. We are given little information about it. If I read things correctly, the predicted protein size is ~325 amino acids (Supp Fig S3A). The colony formation PCR component (Supp Fig 9B) appears to be ~282 bp (which would be ~94 amino acids). Is the PCR meant to show merely the presence of the insert, or does it define the full insert? If a full insert, then the polypeptide/protein piece is ~10kDa, which looks to be in the vicinity of the purified protein coupled to GST (Fig 7). If this is proprietary, that statement can be made, but there does need to be clarification as to whether this sequence was chosen due to algorithm-predicted B cell (or T cell) antigenicity, and to whether any putative post-translational modifications might be missing due to bacterial production of the antigen.

July 28th, 2025

Dear Editor,

EMBO Journal,

The enclosed manuscript entitled “**Arthropod exosomal cement protein as a potential vaccine candidate effectively reduces tick blood-feeding and pathogen transmission**” by Drs. Waqas Ahmed, Wenshuo Zhou, Md. Bayzid, Denae Nadine LoBato, Kehinde D. Fasae and Profs. Girish Neelakanta, and Hameeda Sultana is resubmitted after revision for consideration in the Journal of “EMBO”. The manuscript has been thoroughly updated as a revision and all the comments or suggestions provided by the three reviewers were considered in this revised version of the updated manuscript. We also thank all the reviewers for their great input. The point-by-point responses to each of the reviewer’s comments are provided in this document as comments (in black) and author responses are indicated in blue font. The suggested changes are presented throughout the revised manuscript in “red” color text in the marked/highlighted version of the .pdf file. Word document of manuscript file and response letter without highlights are also included in this resubmission for further processing.

We thank the editor(s) and all the reviewers for providing good feedback to improve this manuscript. In this revised version of the manuscript, we have made the changes throughout the manuscript and supplemental information file (shown as red font) and have responded to all the comments and from all the reviewers. Please find our point-by-point response (shown as blue font) for the comments below.

Referee #1:

The manuscript "Arthropod exosomal cement protein as a potential vaccine candidate effectively reduces tick blood-feeding and pathogen transmission" describes target validation of several Ixodes scapularis tick proteins in feeding and transmission of the Langat virus by ticks. Overall, the manuscript is written with clarity, and it is difficult to follow. The central objective of this study is not apparent and makes it difficult to understand the basis of this work. Significantly, weaving the word "cement" in this manuscript is not based on evidence or background data in literature. It is unclear why proteins in this study were referred to as cement proteins as the authors provide no evidence that the proteins in this manuscript were involved with tick cement formation. Notably, the publication (PMID: 36494396) that described proteins that form Ixodes scapularis cement was not cited in this study demonstrating that the review of literature on tick cement proteins was not well done. Furthermore, the authors mention throughout the manuscript that the proteins are heat shock proteins; however, XM_002400035 is indicated as a glycine rich protein, please verify your information.

We thank the reviewer for the comment about the manuscript clarity and since there is a large amount of data in this paper it could be followed easily as step-by-step readout. We have presented all the data in a very logical fashion so that it is easily understandable. The central objective of this study is to investigate the tick exosomal salivary cargo that could be responsible for the delay in wound closure,

and repair process shown in our previous publication (Zhou-----Sultana, et al, *Frontiers in Cell & Developmental Biol*, 2020). We have added few lines (in introduction) about the central objective of this study and why it is important to understand the basis of this work. We have combed the *Ixodes scapularis* genome and have collected all the NCBI accession numbers (based on the NCBI nomenclature for of these molecules) that have been designated as cement or cement like molecules. Thus, weaving the word as “cement” is purely based on their accession numbers and the details or information provided on the NCBI or VectorBase websites. There is redundancy in cement family of proteins where some of the cement proteins have been shown in the literature to play roles in tick cement cone formation, however, this was not the scope of our study. Our study was designed to explore the cargo molecules in the tick salivary exosomal content. We are sorry that we did not include the vast amount of literature presented in cement field, but we have now included this publication (PMID: 36494396) as per the suggestion from the reviewer. In addition, we have included other literature on tick cement proteins that is relevant to this work. We are sorry for any confusion with the reviewer about the heat-shock proteins and cement proteins and these are two different classes of proteins, and we have not mentioned anywhere about such correlation. XM_002400035 is a glycine rich protein, and several of the cement proteins in this family are similar with glycine rich region between the amino acids 185-272 (please see Supplementary Figure 1).

Additional queries/suggestions to improve the manuscript are below.

Title. The title is wrong, there is no evidence that the proteins studied in this study are cement proteins, please revise.

We respectfully disagree with the reviewer that the title is wrong. We have selected all these genes and protein sequences from NCBI and VectorBase (very well-known websites) that directly calls out these molecules as cement family members. We are going with the wording from public databases. Molecules from the same family could play various roles and independent of one another. For example, Semaphorin 7A, a neuronal guidance factor acts an immune regulator and connects the immune-nervous system, and none of the other Sema proteins plays such unique function. Our future research will enhance the role of these molecules in cement cone formation.

Abstract. The abstract is too descriptive, authors could consider highlighting in the abstract major outcomes in this study.

We have summarized and highlighted all the major results of the study in the abstract. We have now also added a sentence on the study objective.

Introduction. The objective for this study is unclear, please provide your reasoning for this study here. Literature review is incomplete,

We have now added the central objective of the study in introduction, also, we have provided the reasoning for this study. We have added additional literature/citations in this study.

Results. The results section is not well written. Given that the materials and methods are in supplemental, authors should provide a couple of sentences on methods used. For instance, on RNAi silencing, what does mock dsRNA mean? This is confusing to me. Did you inject non-tick related dsRNA, this is what is required for a control. Please specify in results. Provide quantitative data. For instance, this sentence "These data implicate that reduced body weights is likely a result of decrease blood-feeding by these cement-knockdown ticks" does not mean much, please state % of folds reduction. Similar descriptive statements must be accompanied by quantitative data.

With several modifications the results section is clearer now. We have read the results section for several times, and we believe that it easily followed with the details and highlights from the study. In addition, we have made sure that the results section is easily understandable. We have now added a brief section about the Methods (brief section in the manuscript) shown in Supplemental information. As control we used the multiple cloning site region from empty pL4440 plasmid as template to synthesize mock-dsRNA control and this is mentioned in both methods section (under dsRNA synthesis, tick cell transfections, tick microinjections and acquisition/transmission studies) and in results section (please see page 12 and line 261. The empty plasmid pL4440 is a non-tick related dsRNA that serves as a

control for XM_002400035-dsRNA. We have specified this in results section. We have also stated the percentage of fold reduction in the results section of the revised manuscript. All data is shown with the respective quantitative analyses.

Figures. Handwritten labels on tick images don't look good, please type.

We have now modified the tick images in Figures 6A and 7D and H without the hand labels. In 7D image, we cannot cut off the letters "ment". Before, sticking the ticks on double sided tape on glass slides, we have pre-labeled the tick groups and hence those images came out with handwritten labels. We have typed the groups.

Referee #2:

Tick-borne diseases are in expansion worldwide with serious risks for human and animal health. The need of vaccines is urgent and the concept of anti-tick vaccine is regularly mentioned as a solution to limit these diseases. Up to now, numerous tick saliva molecules have been identified but none have been shown to have a real protective effect to stop the transmission of pathogens.

We thank the reviewer for a clear description of the literature.

The manuscript by Ahmed et al. describes the identification of a cement protein from tick saliva involved in the transmission of Langkat virus. Among seven cement proteins deposited in genbank, the authors selected one particularly involved in the process of *Ixodes scapularis* blood feeding. The Langkat virus specifically upregulates this protein in exosomes released during the tick bite and uses these exosomes to facilitate their transmission.

Using different techniques of RNAi silencing, they show that the virus interacts with this protein in tick cell lines and in mouse model. The silencing of the cement protein reduces the virus transmission. They propose this cement protein of tick saliva as a potential vaccine candidate to reduce the tick blood meal but also to reduce the virus transmission. All the project is well structured with numerous experiments to validate their hypotheses, and thoroughly documented. They conducted different techniques in vitro and in vivo to demonstrate the major role of this cement protein.

We thank the reviewer 2 for the detailed description of this study. Also, we are very thankful to the reviewer for comments "All the project is well structured with numerous experiments to validate their hypotheses and thoroughly documented. They conducted different techniques in vitro and in vivo to demonstrate the major role of this cement protein".

Few comments on the study:

However, the concept of an anti-tick vaccine based on a tick saliva protein isolated from cement is not new. Labuda and Nuttall have already demonstrated the role of cement protein in the transmission of tick-borne encephalitis virus, and have proposed testing a specific cement protein as a vaccine to block the transmission of tick-borne encephalitis virus (Labuda et al. Plos Pathogens 2006 - cited by the author). More recently, Fikrig's group at Yale has also proposed testing the cement protein as a candidate vaccine (Lynn et al. TTBDs 2022). Numerous tick saliva proteins have already been identified in different ticks, but none has really shown any effect in blocking transmission of infectious agents. At most, the tick's blood meal is reduced (as shown here) and the female lays fewer eggs, or an allergic reaction is induced by immunization with the saliva protein, leading to rejection of the tick.

We agree with the reviewer 2 that the concept of anti-tick vaccine based on a tick salivary protein isolated from cement is not new, but our cement protein turns out to be a very unique candidate. This cement protein present in tick salivary exosomes has nicely been secured and protected. We believe that being a salivary exosomal molecule, we can propose this molecule as a potential anti-tick vaccine candidate. We are very much aware of all the literature from Drs. Nuttall and Fikrig groups. We have great respect for all their and others including Drs. Mulenga and Karim's published work. Our work highlights a novel tick cement protein that is an exosomal protein and no such work is described in the literature about a tick salivary exosomal cement protein. We agree that "Numerous tick saliva proteins have already been identified in different ticks, but none has really shown any effect in blocking transmission of infectious agents." But our exosomal cement protein is novel and we may see great

effects in the future studies. Our immediate future studies will look at generation of antibody against the tick salivary exosomal cement protein and perform passive immunization studies and further characterization of this molecule.

- I am surprised that none of the 7 proteins is clearly identified as a precise protein. Different studies characterized different tick cement proteins in different tick genera. Both NCBI (protein and nucleotide accession numbers) and VectorBase (accession numbers) websites refer these accession numbers as cement molecules. We do believe that all these molecules have some relation to one another. Please refer to our Supplemental Table S1 for the accession numbers. The accession number XM_002400035 for tick exosomal molecule is still referred as *Ixodes scapularis* cement protein, putative, mRNA (189 bp). The GenBank protein accession number EEC20123.1, also calls it out as cement protein, putative [*Ixodes scapularis*] with residues 1-62 amino acids. We have performed ClustalW alignment and calculated percentage of identity that showed that these molecules are related to each other. The sequence comparison also showed that these are glycine rich proteins and are like the other well-studied cement proteins.

- Line 27-28 : "The molecular mechanism(s) for cement secretion at feeding pit and their role in tick-pathogen interactions is not clearly understood". I do not agree. The cement has been very well characterized and its function is clearly established.

We have modified this statement with exosomal, and hence it becomes clear.

- Figure 4. How do the authors explain that the cement protein they identified is not upregulated in male and particularly female ticks? The blood meal of female lasts for several days and cement is essential to anchor the ticks to the skin.

Our immediate future studies will address the role of tick salivary exosomal cement protein in blood feeding and how it can be variable in male and female adult ticks. Our data from unfed male and female adult ticks (Figure 4A) showed no differences but exploring these ideas will be in our future list of do experiments.

- viral infestation of larvae or nymphs: is immersing nymphs in a viral suspension really effective for infecting ticks? I have a few reservations. I've never really trusted this type of infestation, it's too artificial.

Dr. Marshall Bloom, our groups and several other peers in the tick community have been doing this synchronous infection of ticks. We have observed 60-70% of infection acquired by this method. However, our current study has used synchronous infection method (for ticks used in Figure 2) and all infection in ticks is obtained upon natural route of infection via blood feeding on ticks.

- I also have some reservations concerning the use of the HACAT cell line ; primary human keratinocytes available commercially would be more reliable. HaCaT cell line is from immortalized human keratinocytes cell line derived from normal human cell line. We have performed several experiments with this commercially available cell line. Also, since our specific aim is to identify the tick exosomal molecule that delays the wound healing and repair process, we preferred the same cell line.

- minor errors:

- o line 483: Replace "Its shown' by "It shows"

Replaced

- o line 489: Replace "it as developmentally regulated" by "it is"

Replaced

- o line 551 : replace "our study highlights an important exosomal" by "our study highlights the role of an important exosomal..."

Replaced

o Line 816: figure legend: "seven cement" genes or molecules ?
Added word genes, thank you.

o Line 829: same as line 816
We have changed the line 816.

Referee #3:

EMBO J

EMBOJ-2024-118127

Ahmed et al

Arthropod exosomal cement protein as a potential vaccine candidate effectively reduces tick blood-feeding and pathogen transmission

This is an interesting, if very dense, manuscript that focuses on an *Ixodes scapularis* tick cement component, cement-XM_002400035 molecule, and its (exosomal) roles in tick bio/pathology (particularly in acquisition and transmission of LGTV), and whether this molecule could be a target for effective immunization. As tick-borne diseases are on the rise worldwide, it is imperative to determine mechanisms of disease transmission with the hopes of determining effective ways to prevent these diseases. As such, the studies are largely well done, but the manuscript is a difficult read. Much of this has to do with following all the life stages and feeding cycles of the ticks, but it begins with the "cement-XM_002400035 molecule" designation. While this is an official term, presumably, it becomes cumbersome as often as it is repeated in the text, particularly in the context of mock and dsRNA inhibition. Could there be a way to shorten it after the first time it is denoted?

We thank the reviewer for all the positive comments and well described summary of this study. We agree that our study is very dense, and we have provided a lot of data to support our findings. We agree that there are several terms, but we have shortened several of them and have used the abbreviated forms. For the dsRNA terminologies, it will be better to described them as it may bring up confusions. Instead of repeating the term of tick exosomal cement protein, we have used the accession numbers throughout the manuscript.

For other areas, please be careful to note when you are measuring or describing transcripts vs protein, especially after you generate the murine antisera that can be used to identify or measure protein. Thank you for the suggestion, we have carefully read the manuscript and have added protein in few places.

Also, there are many instances of need for articles ("a", "an", "the") and subject-verb agreement, particularly in singular or plural forms. I have identified some of those, but then lost further energy to do so. I will leave the rest to the authors and copy editors.

All authors have carefully read the manuscript and have considered some minor changes with grammar. We believe that we have removed "the" from not needed places.

Comments follow in order of the manuscript.

Intro

Line 52: the *and* between tick species names probably does not need to be italicized.
Thank you for pointing this out, we have removed the italics on word "and".

Lines 54-55: it is unclear if "Langat Virus-LGTV," is the model pathogen similar to TBEV/POWV (it does not seem to be?), or what is the model pathogen?

We have modified this and have added a parenthesis to show LGTV as the model pathogen.

Line 63: *in* should be deleted.

Removed

Lines 78-79: you may wish to state that 64TRP as an antigenic vaccine component shows the ability to control tick-borne diseases.

We have modified this sentence as per the suggestion.

Line 80: I think it would be clearer to remove *establish to* from the sentence.

Removed

Line 111: *chemokine* CXCL-12 should be singular.

Changed

Supp M&M etc

Line 54: perhaps should say "Uninfected ticks, to be used as controls, were fed on naïve C57BL/6 mice."

Changed

Line 83: *detailed* information....

Changed

Lines 105-106: need to mention that those filters are for the Spectradyne instrument, as this follows right after Sartorius membrane filtering.

Added

Line 134: *supernatants* (plural)

Corrected

Lines 158-159: there are two "obtained from" mentions; it appears it is only Boster Bio that is the source?

Thank you for this point out, we have now modified this sentence.

Line 170: *assays* were performed

Changed

Line 204: "...sites...*were* individually analyzed...."

Changed

Lines 285-292: this might be easier to read as a bulleted list; there is a lot going on, here.

We have now categorized these samples as a-d, for ease.

Results

Bioinformatics; Supp Fig S3 - it might be helpful to highlight (say, in yellow, or bold font) EEC20123.1 in the axes locations, or direct the reader to it saying it is on the far left of each graph in Supp Fig S3
We thank the reviewer for this suggestion, we have now boxed the bar in the each of the graph to highlight EEC20123.1 protein.

Results Section: LGTV-infection selectively upregulates expression of cement-XM_002400035 in tick cells/tick cell-derived exosomes and enhances exosome secretion

Line 180+: Regarding Fig 3 and Supp Fig S5, what is the justification for using actin message for normalization of cement component messages in exosomes? Would it not be better to use particle count or mass of exosomes, as it is not clear in the EV field that any particular protein or message is universally equally loaded into exosomes/EVs? Second, as mRNAs in exosomes are often fragments,

is this full-length cement-XM_002400035 (or other cement messages)? Third, would it not be more important to look for the protein in exosomes instead of mRNA, ie, what is the justification for looking at the message rather than the protein product? I realize that protein detection may be difficult, and it is somewhat shown in Fig 7C (not the greatest blot), but in the biology of cement components in exosomes, what do the transcripts actually do? Finally, there needs to be some mention, perhaps in the legend, in Supp Fig S5 that the quantities (y-axes) are quite different, in some cases, several orders of magnitude (see Sig S5 E vs F, or even vs G). Thus, while XM_002434249 may not change significantly in cells upon infection, there is far more message under all conditions compared to XM_002400035.

The data shown in Fig 3 is for cement-XM_002400035 transcripts in both tick cells and in exosomes. All the panels or data shown in Supplemental Figure 5 is transcript loads of all seven cement transcripts expression in tick cells. 1) We do detect the actin transcripts and with no changes between uninfected and LGTV infected groups, we have normalized our cement-XM_002400035 transcript levels and LGTV PrM gene transcript levels to actin transcripts. Also, actin transcripts are highly expressed in tick exosomes. This is the justification to normalize to cement-XM_002400035 transcripts to actin. The transcripts of actin are used as highest expressed housekeeping gene in tick cells.

In exosomes or in cells, we cannot rely on protein or message and their universal expression, but we are using equal amount of cDNA (and from the same batch of preparation) to analyze the actin transcripts and cement genes. 2) For QRT-PCR analysis, we used an amplified gene fragment of cement-XM_002400035 transcript (nearly 100 bp) and it is not a full-length gene product. This remains same for all other cement genes described in this study. Please refer to Supplementary Figure 4, showing the amplified gene fragments used for generating standards for QRT-PCR analysis. 3) Cement proteins are found only in tick and perhaps other arthropods, and there are no mammalian orthologs detected yet in higher systems. Hence, there are no commercial antibodies available for our use to perform immunoblotting analysis. Our immediate future studies will be on generation of antibody against the cement-XM_002400035 protein. The immunoblot shown in Figure 7C, is performed with the immune sera collected from GST-cement immunized mice and therefore the blots are not to the greatest, due to background from immune serum. We would explore the thought on "biology of cement components in exosomes" as our next step but it is not in the scope of this current study. We have updated the Figure S5 legend. Other than XM_002434249 and XM_002414753, all other cement gene transcripts show the same X-axes. However, both XM_002434249 and XM_002414753 transcripts had no changes with LGTV infection.

For the exosome size measurements, it seems as if the average diameters are smaller for exosomes from infected cells, although this can be difficult to discern at a glance. Is it possible at that increased exosomes numbers from the infected cells could be due to release of LGTV itself, or is the virus too small to be detectable with the Spectradyne equipment?

We are using the filters/cartridges for size 65-400 nm (these are the lowest size available from the vendor) and our flaviviruses including LGTV are 40-60 nm in size. Our previous publication (Zhou---Sultana et al., PLoS Pathogens, 2018), has shown that LGTV infection in tick cells release increased number of exosomes of sizes 50-100 nm (with manual counts). We believe that LGTV full-length RNA genomes are perhaps present in these tick exosomes (collected at 72 h post infection).

Section: RNA interference of cement-XM_002400035 impacts its expression and LGTV-loads in tick cells/exosomes and in nymphal ticks during pathogen acquisition

Supp Fig S7: I think the word you mean is "elution", not "elusion".

Thank you for pointing this out, we have corrected it in Figure S7, and its legend.

Fig Legend 5: "Total protein profiles are stain-free gel images". Are those not actually *stained* gel images?

All total protein profile gel images are stained free gels that were either obtained from vendor (Bio-Rad) or pre-casted in our laboratory using the TCE. "TCE gel stain refers to a method of visualizing proteins in polyacrylamide gels using 2,2,2-Trichloroethanol (TCE). It's a method that can be incorporated into

the gel during its preparation, allowing for rapid and sensitive protein detection through UV-induced fluorescence". We have now added this information in methods.

I have the same comments (from above) regarding the loss of viral replication or shedding as part of the reduction in exosome numbers (and loss of genetic and protein components seen in Fig 5). We have shown that LGTV infection increases the exosome concentrations and numbers (please refer to Figure 3), in comparison to the uninfected group. We have noted that upon silencing of exosomal cement molecule (with reduced transcripts in cells and exosomes, please refer to Figure 5), LGTV loads were reduced in tick cells and in exosomes. The observation that LGTV increases exosomes concentration and numbers is directly correlated with the increased viral replication that is regulated by the cement molecule.

For Supp Table S3, was actin ever accounted for in the densitometry?

Actin was not accounted for densitometric analysis as we do not see any differences in the protein loads for actin. We have now added this data into Table S3.

Line 313 and Fig 6E; you may wish to lighten the blot for NS1 as it is not easy to see the differences between mock and dsRNA-treated ticks with this image.

We have now increased the brightness and reduced the contrast for this NS1 blot in Figure 6E.

Line 315: ..."this *immunoblot*..." [singular]

Removed "s" from immunoblots.

In the Fig 6 Legend, lines 954-955, should be "A group of repleted ticks *was* also immediately weighed...."

Replaced were with "was".

Again, as mentioned above, "stain-free" does not make sense, do you mean "destained"?

Fig 6 legend line 974, you do mean panels *6* F-H?

Please see the above explanation about the stain free gel images. We have corrected the number 6 for 7, thank you.

Line 323: Should say "We also observed reduced viral loads in mice tissues (brains and spleens) *from mice* that allowed feeding...." Otherwise, it sounds like the reduced viral loads allowed feeding of ticks.

Modified.

Line 327: *treated*

Corrected, thank you.

Supplemental section, legend to Supp Fig S10, line 592, should just be *engorge and were collected* Supp Fig S11, legend, lines 616-617, the terms "Mock-I" and "dsRNA-I" never show up in the figure.

Corrected.

Results Section: Immunization with exosomal cement protein impairs LGTV-transmission from infected ticks to naïve vertebrate host

Line 340: *bind*

corrected

Line 344: presumably these are "mice serum *antibodies*..."

We have added "antibodies".

Lines 371-375, while this is in the Supp Materials and Methods, the detection of anti-cement Abs in the nymphs/adults deserves a little better technical explanation here in the results, because this is rather fascinating. Here, GST or GST-cement proteins were used as capture substrates for Abs in the tick

lysates, followed by anti-mouse Abs for detection. Given that you were using the mouse sera as a probing agent for most of these assays, this is a necessary clarification.

We agreed with the reviewer and have now explained this results in detail. We have also added a discussion point for this data in the revised version of the manuscript. Also, we have updated the methods section for details on this experimental set up. Yes, we coated the wells with purified proteins as substrates for antibodies present in tick lysates (described above) and probed the infected tick lysates from fed nymphs or molted adult ticks (collected from GST/GST-cement-immunized mice) and followed by detection with anti-mouse HRP conjugated secondary antibodies.

Figure 7 legend:

Line 991: *are* shown

Corrected

Lines 991-993: This is a difficult sentence given the number of / in it and the number of immunoblot targets. Also, it should be "cement-XM_002400035-*dsRNA*."

We have modified this sentence and have corrected dsRNA.

Line 996: "A group of ticks *was* ..weighed..." (In this case you specifically stated a single group).

Corrected

Lines 1008+: you begin referring to panels "10B" etc through here - I presume you mean panel 7.

Thank you for the point out, we have now corrected this in the revised version of manuscript.

Results Section: Silencing of exosomal cement reduced inflammation at tick bite site in mice and exogenous GST-CXCL-12 treatment closed wounds in human keratinocytes

There needs to be better explanation of Fig 8 and Supp Fig S12, in particular, the arrows/arrowheads are too small, and need have better distinguishing features (particularly in the denser cellular backgrounds). For the wound/scratch assay, there needs to be some explanation of the expected results vs what is seen (or in conjunction with what is seen), because there are multiple levels of complexity inherent in the experiment. Are there any statistics applied to these measurements? In Fig 8E, what is GST-C6?

We have clarified all the points and have better explained this part of the results. We have now described the wound/Scratch assays. Statistical data for the wound assay is shown in panel 8E, and GST-C6, indicated GST-cement protein group. We have now changed this label in the revised version of the manuscript.

Figure 8 legend

Lines 1037-1038: LGTV-infected and mock-dsRNA-treated (M) or LGTV-infected and cement-XM_002400035-dsRNA-treated groups *are* shown

Corrected

We do not see "M" anywhere in the figure?

Removed "M"

Line 1058: this is just a question - would Langerhans cells (ie, epidermal DCs) and dermal DCs have any impact on tick feeding? These would seem more immediately responsive than circulating blood DCs. Just wondering.

These are the wonderful thoughts, we have not tested the Langerhans cells (ie, epidermal DCs) and dermal DCs impact on tick blood feeding.

Line 1059: "tick cement exaggerates residential keratinocytes..." to do what?

We have corrected this sentence for better understanding.

Discussion

Line 483: *It has been shown that*

Corrected

Lines 485-86: Please be clear here, and in general, when you say "It is intriguing to note that only cement-XM_002400035 is upregulated in LGTV-infected tick cells and in exosomes" what you refer to - protein, or transcript? Since you have potential antisera now, it will not be inherently obvious when you mean transcript vs protein.

We have modified this sentence.

Paragraph starting at line 522: the uptake of antibodies from host to ticks is fascinating, but likely requires high titer of Ab. Is there anything known about how much Ab may be "retrotransmitted" to ticks by any given animal, much less an immunized one? That would be an important determinant in further pathogen transmission/inhibition outside the host. While beyond the scope of this already dense manuscript, it could also conceivably be quantified in future experiments where quantity of Ab transmission to ticks is compared to Ab titer in the host.

We thank the reviewer for such thoughtful discussion of ideas. We have included a sentence to enhance the impact of this work with future studies. Antibodies produced in murine host could be determined and one can determine the quality and titers of these antibodies. However, how much of these antibodies (their quality, titers and longevity) are ingested via blood meal into ticks is challenging but achievable. Surely, our future research will step towards the determination of answers for some of these interesting thoughts.

Lines 538-9: "Exogenous treatment of GST-cement-XM_002400035 delayed wound closures." A truly interesting experiment here would be to show that anti-cement antisera could alleviate the cement-induced wound closure delay. Do you have enough material remaining to do that?

We really liked this challenging experiment proposed by the reviewer that surely did enhance our study. Unfortunately, we did not have sufficient GST-cement purified protein to perform this experiment. We were very much inspired by this idea and freshly generated the GST-cement purified protein, and immunized mice to collect serum from GST-cement protein immunized mice and GST-protein immunized mice. We performed a new wound healing/scratch-based assay with untreated (UT), Cement protein alone, GST-protein immunized mice serum & cement protein and GST-cement protein immunized mice serum & cement protein. **We found that cement protein in the presence of anti-cement antisera was ineffective in delaying wound healing compared to the controls.** This data was very interesting to the current manuscript and has been added as a new Figure 8 in the revised version of the manuscript. The previous Figure 8 is moved back as Figure 9.

One limitation must be stated in this study, and that is for experiments involving murine antisera and GST as a target, the mice from this study did indeed have Abs vs GST (as well as the cement Ag; Fig 7C). Thus, there could be background reactivity to the GST component. For future experiments, it might be suggested to use a cleavable GST linker to remove that following purification and prior to immunization.

We have performed several other studies with GST protein immunization in mice, and we have not seen any background reactivity to the GST component. We thank the reviewer for this suggestion to use the cleavable GST linker. Our GST-protein cloning vector pGEX-6P-2 comes with the PreScission cleavage site that allows the cleavage at low temps. We have added this information in the methods section in supplemental information.

There should also be some discussion as to the choice of cement fragment/polypeptide used for immunization, ie, the putative protein sequence should be shown. We are given little information about it. If I read things correctly, the predicted protein size is ~325 amino acids (Supp Fig S3A). The colony formation PCR component (Supp Fig 9B) appears to be ~282 bp (which would be ~94 amino acids). Is the PCR meant to show merely the presence of the insert, or does it define the full insert? If a full insert,

then the polypeptide/protein piece is ~10kDa, which looks to be in the vicinity of the purified protein coupled to GST (Fig 7).

If this is proprietary, that statement can be made, but there does need to be clarification as to whether this sequence was chosen due to algorithm-predicted B cell (or T cell) antigenicity, and to whether any putative post-translational modifications might be missing due to bacterial production of the antigen.

We have now added this information on the selection of the cement protein polypeptide or fragment (35 amino acids or 105 bp fragment) used for the immunization studies in results and supplemental details (cloning section). Please see the new Supplemental Figure 9A. The predicted protein size of tick exosomal cement protein is 62 amino acids (please refer to supplemental Figure 9A and <https://www.ncbi.nlm.nih.gov/protein/215510670>) for the complete sequence information. Also, please refer to GenBank: EEC20123.1 (for protein sequence information) or nucleotide accession number XM_002400035. The PCR is meant to show the presence of insert, and the polypeptide or protein size is ~10 kDa, and coupling to GST protein (26 kDa) will lead to a protein product size of 36 kDa of this GST-cement protein (as shown in Figure 7A). We have now added the marker details and some of this information on the Figure 7 and supplemental information. We have not claimed it yet to be proprietary and would not like to make any comments on this regard. We do not assume that any putative post-translational modifications might be missing due to bacterial production of this antigen. We have cloned expressed and purified several of the tick proteins and have not seen any issues with our system.

Thanking you,

Sincerely,

Hameeda Sultana, Ph.D.,

Dear Prof. Sultana,

Thank you for submitting your manuscript for consideration by the EMBO Journal. It has now been seen by three referees whose comments are enclosed. As you will see, all three referees express interest in your manuscript and are broadly in favour of publication, pending satisfactory minor revision.

Please also make sure to take care of the technical issues raised by our editorial assistance team, which are provided below.

Given the referees' positive recommendations, I would like to invite you to submit a revised version of the manuscript, addressing the remaining comments of the reviewers.

We generally allow three months as standard revision time. As a matter of policy, competing manuscripts published during this period will not negatively impact on our assessment of the conceptual advance presented by your study. However, we request that you contact the editor as soon as possible upon publication of any related work, to discuss how to proceed.

Thank you for the opportunity to consider your work for publication. I look forward to your revision.

Yours sincerely,

Yehu Moran
Editor
The EMBO Journal

We realize that it is difficult to revise to a specific deadline. In the interest of protecting the conceptual advance provided by the work, we recommend a revision within 3 months (16th Dec 2025). Please discuss the revision progress ahead of this time with the editor if you require more time to complete the revisions.

editorial assistance team comments*

*AUTHOR CHECKLIST: missing - Please provide a completed Author Checklist, which you can download here:
<https://www.embopress.org/pb-assets/embo-site/EMBO%20Press%20Author%20Checklist-1642513524327.xlsx>

*Keywords: 11 - Please reduce the number of keywords to a maximum of 5

*Author Contributions: Please remove the author contributions from the manuscript text and ensure that they are comprehensively listed in our system using the CRedIt format.

*DisclCIS: Please rename the corresponding section "Disclosure and Completing Interests Statement"

*APPENDIX 1 FILE WITH ToC: Please rename the PDF with the supplemental materials "Appendix", remove the supplemental methods from the PDF file and add them to the main manuscript. Please rename the supplemental figures "Appendix Figure S1" etc. in the PDF and in the manuscript text. The correct nomenclature of the supplemental tables is "Appendix Table S1" - S4. Please add a table of contents to the appendix and include page numbers. The line numbers in the appendix file can be removed.

*REAGENT TABLE: Please upload a reagents and tools table (which you can download from)

https://www.embopress.org/pb%2Dassets/embo-site/Reagents_Tools_Table_TEMPLATE.docx as a separate file.

*SYNOPSIS IMAGE: Please upload the image in jpg or png format and sized exactly 550 pixels wide x 300 - 600 pixels high

*SYNOPSIS TEXT: not provided. Please provide.

DATA CHECK: PASS

- Figure legends:

Please note that the exact p values are not provided in the legends of figures 2G, 3A, B; 4A, 6B. Please provide.

Figures

Figure reuse between Figure 9 A & B with Appendix Fig S12 A & B.

As the authors say this is intentional and added the info to legend of Fig. 9, they should also add this to legend of Appendix Figure S12.

Referee Reports

Referee #1:

The revised manuscript shows substantial improvement. However, a major concern remains: the authors claim to have characterized tick cement proteins, yet no evidence supporting this is presented. As a result, the manuscript title is misleading.

1) Evidence or renaming of proteins

Unless the authors can provide clear evidence that the proteins described in this manuscript are present in tick cement, I recommend the following:

1A) Rename the proteins accordingly. Instead of referring to them as cement proteins, use a neutral or descriptive nomenclature based on primary sequence features (e.g., glycine-rich, heat shock, or other appropriate classifications) or on computational secondary structure predictions.

1B) Revise the manuscript title to remove the word cement. Retaining this term without evidence is fundamentally misleading, as not every glycine-rich or heat shock protein is necessarily a component of tick cement. If the authors wish to continue referring to these proteins as "cement" proteins, they must conduct an experiment demonstrating their presence in tick cement.

2) Accession numbers

When referencing proteins, please provide the appropriate protein accession numbers (e.g., XP₁). The accession number XM_002400035 refers to nucleic acid, not protein.

Referee #2:

Dear authors,
Dear editor,

The authors responded to my comments. However, I remain disappointed that the suggestion to test primary human keratinocytes was not tested. Most cell lines are unreliable, particularly Hacat cells.
A one-year review would have allowed this test to be carried out.

Referee #3:

The authors have responded well to my questions and comments.

Dec 19th, 2025

Dear Editor,

EMBO Journal,

The enclosed manuscript entitled “**Arthropod exosomal glycine-rich protein as a potential vaccine candidate effectively reduces tick blood-feeding and pathogen transmission**” by Drs. Waqas Ahmed, Wenshuo Zhou, Md. Bayzid, Denaé Nadine LoBato, Kehinde D. Fasae and Profs. Girish Neelakanta, and Hameeda Sultana is resubmitted after revision for consideration into the Journal of “EMBO”. The manuscript has been thoroughly updated as a revision and all the comments or suggestions provided by the editors and three reviewers were considered in this revised version of the updated manuscript. We also thank all the reviewers for their great input. The point-by-point response to each of the reviewer’s comments are provided in this document as comments (in black) and author responses are indicated in blue font. The suggested changes are presented throughout the revised manuscript in “red” color text in the marked/highlighted version of the .pdf file. Word document of the revised manuscript file without highlights is also included in this resubmission for further processing.

We thank the editor(s) and all three reviewers for providing good feedback to improve this manuscript. In this revised version, we have made the changes throughout the manuscript and supplemental or Appendix information file (as red font) and have seriously responded to all the comments from the reviewers. Please find our point-by-point response (shown as blue font) for the comments below.

Referee #1:

The revised manuscript shows substantial improvement. However, a major concern remains: the authors claim to have characterized tick cement proteins, yet no evidence supporting this is presented. As a result, the manuscript title is misleading.

1) Evidence or renaming of proteins

Unless the authors can provide clear evidence that the proteins described in this manuscript are present in tick cement, I recommend the following:

1A) Rename the proteins accordingly. Instead of referring to them as cement proteins, use a neutral or descriptive nomenclature based on primary sequence features (e.g., glycine-rich, heat shock, or other appropriate classifications) or on computational secondary structure predictions.

1B) Revise the manuscript title to remove the word cement. Retaining this term without evidence is fundamentally misleading, as not every glycine-rich or heat shock protein is necessarily a component of tick cement. If the authors wish to continue referring to these proteins as "cement" proteins, they must conduct an experiment demonstrating their presence in tick cement.

We thank this reviewer 1, and we have added glycine-rich label to this protein.

As per our previous response in first revision, we had combed the *Ixodes scapularis* genome and collected all the NCBI accession numbers (based on the NCBI nomenclature for these molecules) that have been designated as cement or cement like molecules on NCBI. Thus, weaving the word as

“cement” is purely based on their accession numbers from NCBI (as nucleotide or proteins) and the details or information provided on the NCBI or VectorBase websites. There is a redundancy in cement family of proteins where some of the cement proteins have been described in the literature to play roles in tick cement cone formation, however, not all cement molecules may play such roles, and this type of finding was not in the scope of our present study. Our study was designed to explore the cargo molecules in the tick salivary exosomal content that could be responsible for the delay in wound closure, and repair process reported in our previous publication (Zhou-----Sultana, et al, *Frontiers in Cell & Developmental Biol*, 2020). Our future work will surely focus on studying the role of this tick exosomal molecule in the tick cement cone. For now, in the current manuscript, we have renamed the cement label with glycine-rich proteins or GRPs, in short. Not just the title, but the entire text, figures, legends and the information in appendix has been renamed with GRPs. We believe that the reviewer is satisfied with this change.

1A) We have now renamed the protein as per the reviewer’s suggestion. Since tick exosomal novel protein is a glycine-rich molecule, the title reflects the rename for this molecule. We have used a neutral or descriptive nomenclature based on their primary sequences with repeated glycine-rich contents.
1B) The word “cement” is now removed from the title, in Figures/Legends and in the revised manuscript. The word cement-XM_002400035 is replaced with XM_002400035 at several places throughout the revised manuscript.

2) Accession numbers

When referencing proteins, please provide the appropriate protein accession numbers (e.g., XP_). The accession number XM_002400035 refers to nucleic acid, not protein.
We have changed the nucleotide accession number to the protein one.

Referee #2:

The authors responded to my comments. However, I remain disappointed that the suggestion to test primary human keratinocytes was not tested. Most cell lines are unreliable, particularly Hacat cells. A one-year review would have allowed this test to be carried out.

We have now tested the primary human keratinocytes in this extensive and novel study. Our first revision took a year as we did freshly prepare the anti-sera from both GST and GST-GRP proteins immunized mice.

Also, it was extremely difficult to culture and grow the primary human keratinocytes (normal, human, adult, Catalog number PCS-200-011, obtained as authenticated culture from American Type Culture Collection (ATCC)). These cells needed growth supplements (that were purchased and used as per the instructions from ATCC) and with enormous care. It took me 2-2.5 months to standardized working with these cells. The researcher who did the assay failed several times to perform the assay which is either that cells will not adhere to multiple well plates or do get contaminated easily. They would not adhere or grow on different T-25 flask or on various well-plates. Anyway, after several weeks of standardization, we finally did succeeded working on these cells and performed the wound healing assay with multiple replicates and as independent experiments. This new data is shown as supplemental information and as Figure S12 (with panels S12A and S12B). Relevant information has been added in methods, results, discussion and figure legend sections of the revised manuscript.

Referee #3:

The authors have responded well to my questions and comments.
We thank the reviewer 3 for all the support and positivity.

Dear Prof. Sultana,

Before we can officially accept your manuscript we would need you to fix the synopsis text. In The EMBO Journal synopsis has a conserved structure you can find in the instructions for authors and you are welcome also to look on recently published papers in EMBO Journal to get clear examples for this structure. The text you provided for the synopsis, unfortunately, does not follow our regular format. Please fix this issue and resubmit it at your earliest convenience.

I look forward to your revision.

Yours sincerely,

Yehu Moran
Academic Editor
The EMBO Journal

Read our guidance for manuscript revisions and related editorial policies: <https://link.springer.com/journal/44318/submission-guidelines#cms-Revised-submissions>

<https://media.springernature.com/original/springer-cms/rest/v1/content/27825798/data/v1>

- a point-by-point response to the referees' comments, with a detailed description of the changes made (as a word file).
- a word file of the manuscript text.
- individual production quality figure files (one file per figure)
- a complete author checklist
- Expanded View files (replacing Supplementary Information)
- a Reagents and Tools Table as part of the Methods section

Please remember: Digital image enhancement is acceptable practice, as long as it accurately represents the original data and conforms to community standards. If a figure has been subjected to significant electronic manipulation, this must be noted in the figure legend or in the 'Methods' section. The editors reserve the right to request original versions of figures and the original images that were used to assemble the figure.

We realize that it is difficult to revise to a specific deadline. In the interest of protecting the conceptual advance provided by the work, we recommend a revision within 3 months (9th Apr 2026). Please discuss the revision progress ahead of this time with the editor if you require more time to complete the revisions.

The authors addressed the remaining editorial issues.

Dear Prof. Sultana,

I am pleased to inform you that your manuscript has been accepted for publication in the EMBO Journal.

You may qualify for financial assistance for your publication charges - either via a Springer Nature fully open access agreement or an EMBO initiative. Check your eligibility: <https://link.springer.com/journal/44318/how-to-publish-with-us>

Yours sincerely,

Yehu Moran
Editor
The EMBO Journal

Please note that it is The EMBO Journal policy for the transcript of the editorial process (containing referee reports and your response letters) to be published as an online supplement to each paper. If you should prefer removal of any referee-only figures included in the point-by-point response(s), e.g. because they may still be used for future publication or because they have been reproduced from published work by others, please do let us know immediately via response email.

More information is available here: <https://link.springer.com/partners/embo-press/editorial-policies#Peer%20review>